# Mechanics and dynamics of translocating MreB filaments on curved membranes

**Felix Wong[1], Ethan C Garner[2,3], Ariel Amir[1]\***

[1]John A Paulson School of Engineering and Applied Sciences, Harvard University, Cambridge, United States; [2]Department of Molecular and Cellular Biology, Harvard University, Cambridge, United States; [3]Center for Systems Biology, Harvard University, Cambridge, United States

**Abstract** MreB is an actin homolog that is essential for coordinating the cell wall synthesis required for the rod shape of many bacteria. Previously we have shown that filaments of MreB bind to the curved membranes of bacteria and translocate in directions determined by principal membrane curvatures to create and reinforce the rod shape (Hussain et al., 2018). Here, in order to understand how MreB filament dynamics affects their cellular distribution, we model how MreB filaments bind and translocate on membranes with different geometries. We find that it is both energetically favorable and robust for filaments to bind and orient along directions of largest membrane curvature. Furthermore, significant localization to different membrane regions results from processive MreB motion in various geometries. These results demonstrate that the in vivo localization of MreB observed in many different experiments, including those examining negative Gaussian curvature, can arise from translocation dynamics alone.

DOI: https://doi.org/10.7554/eLife.40472.001

**\*For correspondence:**
arielamir@seas.harvard.edu

**Competing interests:** The authors declare that no competing interests exist.

## Introduction

The role of membrane curvature in influencing the cellular location and function of proteins has been increasingly appreciated (*McMahon and Gallop, 2005*; *Zimmerberg and Kozlov, 2006*; *Baumgart et al., 2011*). In cells, membrane curvature can both be induced by proteins that bind to membranes as well as recruit proteins that bind to this curvature (*Peter et al., 2004*; *Renner and Weibel, 2011*; *Wang and Wingreen, 2013*; *Ursell et al., 2014*; *Hussain et al., 2018*; *Wu et al., 2018*). Here, we focus on MreB, an actin homolog essential for coordinating the cell wall synthesis required for the rod shape of many bacteria (*Jones et al., 2001*). We examine how MreB orientation and motion along directions of principal curvature affect its localization in different cellular geometries. MreB filaments polymerize onto membranes (*Figure 1A*) (*Salje et al., 2011*; *van den Ent et al., 2014*; *Hussain et al., 2018*), creating short filaments that move along their lengths in live cells. Viewed dynamically, MreB filaments are seen to rotate around the rod width, a motion powered by the activity of associated cell wall synthesis enzymes (*Figure 1B* and *Figure 1—video 1*) (*Garner et al., 2011*; *Domínguez-Escobar et al., 2011*; *van Teeffelen et al., 2011*; *Reimold et al., 2013*; *Hussain et al., 2018*). The orientation of MreB filaments coincides with the direction of their motion, as filaments move along the direction in which they point (*Figure 1C*) (*Hussain et al., 2018*; *Olshausen et al., 2013*). However, studies examining the localization of MreB in kymographs (*Ursell et al., 2014*) or at single time points (*Bratton et al., 2018*) have found that, in *Escherichia coli*, MreB filaments are enriched at regions of negative Gaussian curvature or small mean curvature (*Figure 1D*). This prompts the question of how the collective motion of MreB filaments could affect, or give rise to, their enrichment. As filaments are constantly moving, their instantaneous localization to any one point is transient: a typical filament (~250 nm long) in *Bacillus subtilis* moves through a 1 $\mu m^2$ region in ~30 s. To understand how the previously observed enrichment at negative Gaussian

curvatures could arise from the dynamics of filaments moving around the cell, we sought to relate the binding and motion of MreB filaments to their distribution in different geometries.

In previous work (*Hussain et al., 2018*), we demonstrated that MreB filaments orient and translocate along directions of largest principal curvature inside differently shaped *B. subtilis* cells and liposomes. To understand how MreB filaments orient along different membrane curvatures and how their motion along this orientation affects their cellular localization, we first model the mechanics of MreB-membrane binding and provide a quantitative description of how MreB filaments bind both in vivo and in vitro. Next, we model the curvature-dependent motion of MreB filaments in different geometries and examine how this motion affects their distribution to different membrane regions.

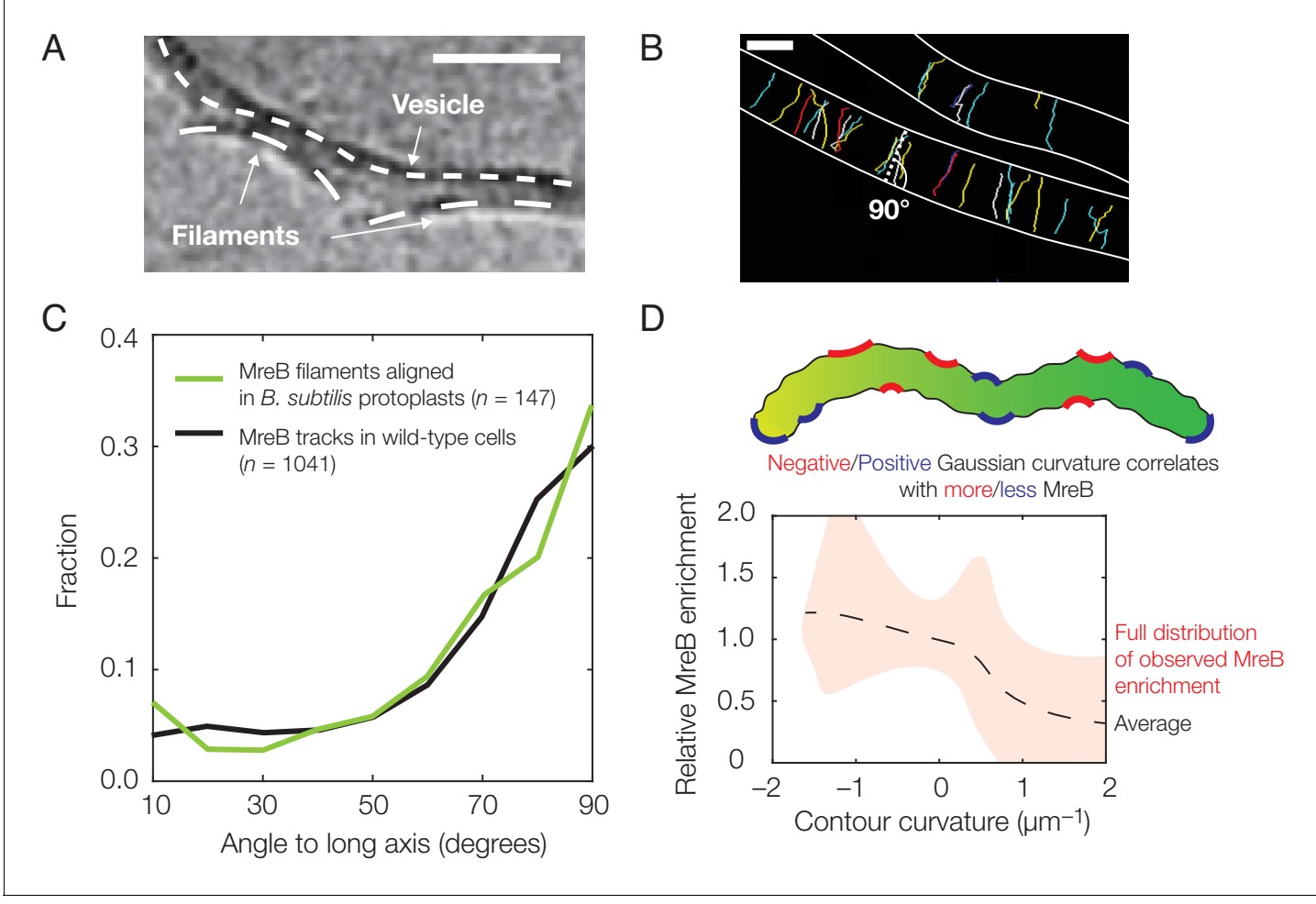

**Figure 1.** Experimental observation of membrane binding and translocation. (**A**) Cryo-electron microscopy image showing the direct membrane binding of MreB filaments reconstituted in vitro to vesicles. The scale bar indicates 50 nm, and dashed curves represent guides. The image is reproduced here from *Salje et al. (2011)* under a CC BY 3.0 license (https://creativecommons.org/licenses/by/3.0/). (**B**) Fluorescence microscopy image of MreB filaments translocating in live *Bacillus subtilis* cells with trajectories of individual filaments drawn and cell edges outlined, reproduced from *Hussain et al. (2018)* (see also *Figure 1—video 1*). Note that typical trajectories are perpendicular to the cellular long axis, consistent with the binding angles in (**C**). The scale bar indicates 1 μm. (**C**) Angular distribution of membrane-bound filaments within *B. subtilis* protoplasts confined to become rod-shaped (green curve) and MreB motion in wild-type *B. subtilis* cells (black curve), reproduced from *Hussain et al. (2018)*. (**D**) Relative MreB enrichment in *Escherichia coli* cells growing in a sinusoidally shaped chamber, adapted from *Ursell et al. (2014)*.

DOI: https://doi.org/10.7554/eLife.40472.002

The following video is available for figure 1:

**Figure 1—video 1.** SIM-TIRF movies of MreB motion.

DOI: https://doi.org/10.7554/eLife.40472.003

Strikingly, we find that the dynamics of MreB translocation alone, without requiring any intrinsic preference of MreB filaments for curved regions of the cell, results in differential enrichment of MreB filaments at regions of negative Gaussian curvature or small mean curvature similar to those observed in cells.

## Results

### Mechanics of binding

We first model how inwardly curved (*Salje et al., 2011*; *van den Ent et al., 2014*; *Hussain et al., 2018*) MreB filaments bind and orient on membranes. Previous theoretical studies have modeled the binding of protein filaments to membranes and demonstrated that binding conformations can be influenced by both filament thickness and twist. In a seminal theoretical work, Wang and Wingreen modeled twisted bundles of MreB approximately six-fold thicker than typical filaments (*Wang and Wingreen, 2013*). This study nicely demonstrated that the mechanics of binding alone could orient bundles and suggested that bundle length could be limited by twist. Another elegant theoretical study by Quint *et al.* demonstrated that twisted filaments of varying rigidities could bind to regions of negative Gaussian curvature in manners that are particularly energetically favorable (*Quint et al., 2016*). While it is intriguing to examine the effects of filament rigidity and twist on general filament systems (discussed below), here we focus on modeling thin, inwardly curved MreB filaments with no twist. We focus on these parameters as they reflect the observations of all available in vitro studies of membrane-associated MreB: three different cryo-electron microscopy studies have shown that membrane-bound MreB filaments are flat and untwisted, binding to the membrane on one filament face (*Figure 1A*) (*Salje et al., 2011*; *van den Ent et al., 2014*; *Hussain et al., 2018*). These studies also suggest that inward curvature, and not twist, limits filament length: MreB filaments are short when polymerized onto non-deforming planar-supported lipid bilayers, but become extremely long when polymerized inside deformable liposomes (*Salje et al., 2011*; *van den Ent et al., 2014*; *Hussain et al., 2018*). The model we present here extends our previous work (*Hussain et al., 2018*) and demonstrates that thin, untwisted filaments orient robustly along directions of largest principal curvature, thus providing a generic mechanism for orienting their motion.

We model an MreB filament as a polymer which binds linearly along its length to a membrane in an energetically favorable manner, namely via burial of hydrophobic residues on one face (*Figure 2A*; see *Figure 2—figure supplement 1* for additional details) (*Salje et al., 2011*). We assume the filament to be a curved, cylindrical, linear-elastic rod which is free to bend to maximize membrane interaction, so that its elastic energy of deformation is $E_{\mathrm{bend}} = (\pi Y r_f^4/8) \times \int (\kappa - \kappa_s)^2 d\ell$, where $r_f$ is the filament cross-sectional radius, $Y$ is the filament Young's modulus, $\kappa$ is the curvature of the deformed state, $\kappa_s$ is the intrinsic filament curvature, and the integration is over the filament length (*Landau and Lifshitz, 1970*). We will minimize the free energy change due to binding with respect to the deformed curvature, $\kappa$.

Next, we assume an isotropic, fluid, bilayer membrane, where there are no in-plane shears and the only in-plane deformations are compressions and expansions (*Safran, 2003*). The membrane free energy is given by the Helfrich form,

$$\mathcal{F} = \int_S \left[ \frac{k_b}{2}(2H - H_s)^2 + \frac{k_t}{2}K + \gamma \right] dA - p \int_S dV, \tag{1}$$

where $k_b$ is the bending rigidity of the membrane, $k_t$ is the saddle-splay modulus of the membrane, $H_s$ is the spontaneous curvature, $\gamma$ is the membrane surface tension, $p$ is the pressure difference across the membrane, $H$ and $K$ are the mean and Gaussian curvatures of the membrane surface, $S$, respectively, and $dA$ and $dV$ denote area and volume elements, respectively (*Helfrich, 1973*; *Safran, 2003*; *Zhong-can and Helfrich, 1987*; *Zhong-can and Helfrich, 1989*). For the small membrane deformations considered in this work, we assume that excess phospholipids can be freely added to the membrane to compensate for stretching, so that the membrane surface tension $\gamma = 0$ (*Safran, 2003*). Note that a nonzero surface tension would make the preferred binding orientation determined below more energetically favorable. For simplicity, we also assume $H_s = 0$ and note that the case of a nonzero $H_s$ can be considered similarly. Finally, we assume that the membrane

surface can be parameterized in the Monge gauge by a function $h = h(x, y)$, where $x$ and $y$ are real numbers and terms of quadratic order or higher in the gradient of $h$, $\nabla h$, are neglected. As shown in Appendix 1, the mechanical energy associated with membrane deformations is determined by the solution of the shape equation

$$\Delta^2 h = \frac{p}{k_b}, \tag{2}$$

which is similar to the equilibrium equation of a thin plate (*Ventsel and Krauthammer, 2001*; *Timoshenko and Woinowsky-Krieger, 1959*). Here $\Delta^2$ is the biharmonic operator and *Equation (2)* is subject to Dirichlet boundary conditions enforcing continuity of mean curvature and surface height in a manner compatible with the deformed filament. *Equation (2)* can then be decomposed as two Poisson equations, each with Dirichlet boundary conditions, and solved numerically using the finite element method (Appendix 1).

The free energy change due to filament binding is determined by $E_{\mathrm{bend}}$, $\mathcal{F}$, and the solution of *Equation (2)*. For characteristic parameter values relevant to the binding of MreB filaments to bacterial membranes, as summarized in *Supplementary file 1*, our model predicts a preferred, circumferential orientation of MreB binding in a rod-shaped cell. This result arises because the intrinsic curvature of filaments is smaller than characteristic cell radii. While the filament bends to conform to the membrane for physiological values of $p$—as shown in previous work (*Hussain et al., 2018*)—and

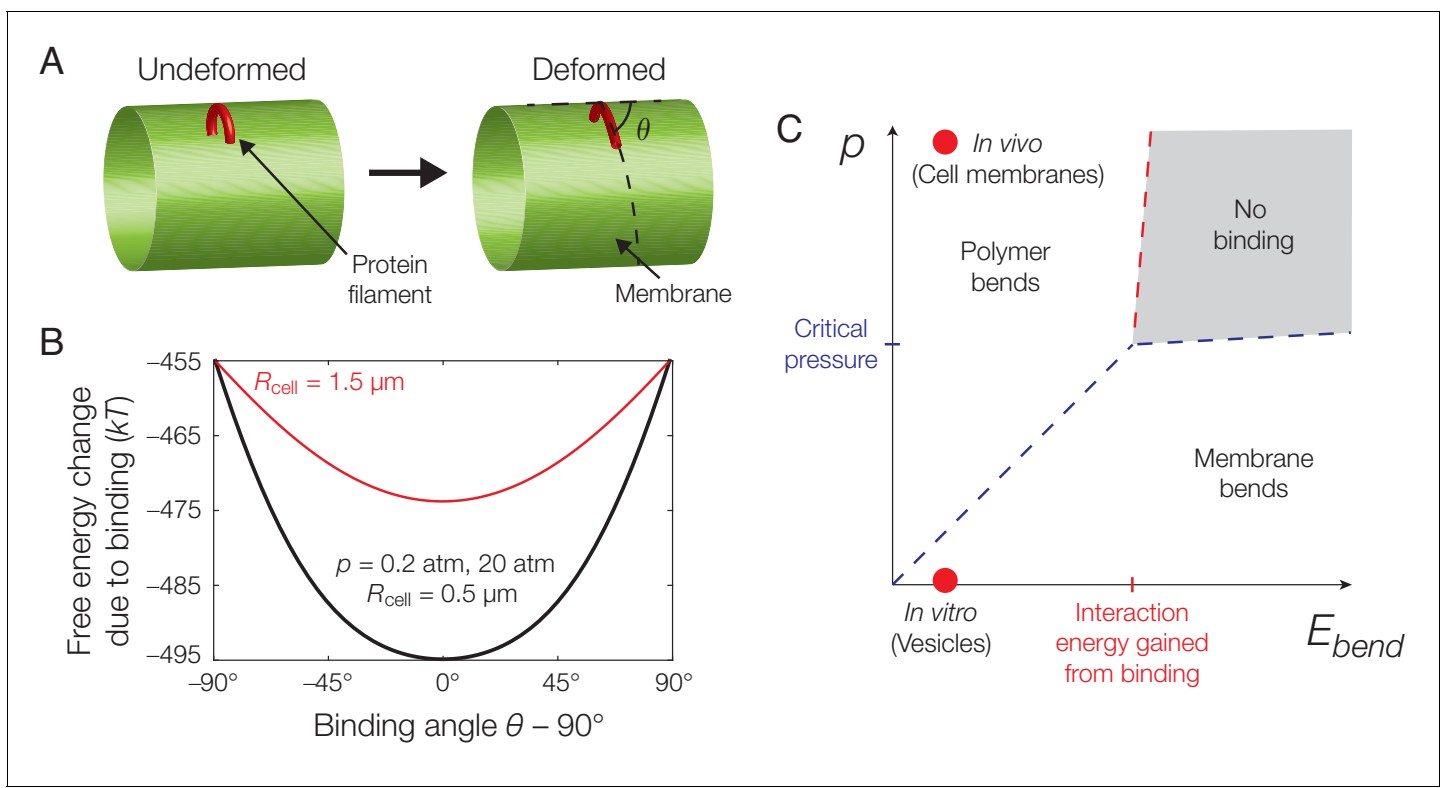

**Figure 2.** Mechanics of binding. (**A**) A schematic of the model. A protein filament (red) may bend and bind to a cylindrical membrane (green) at an angle with respect to the long axis, $\theta$, and the equilibrium conformation may involve deformations of both protein and membrane. (**B**) Plot of the estimated free energy change due to filament binding against the binding angle, $\theta$, where the cell radius $R_{\mathrm{cell}} = 0.5$ $\mu$m, the pressure difference across the membrane $p = 20$ atm, the estimated turgor pressure of *B. subtilis* (*Whatmore and Reed, 1990*), and the other model parameters are given in *Supplementary file 1*. The estimate is similar for $p = 0.2$ atm, but exhibits a more shallow potential well for $R_{\mathrm{cell}} = 1.5$ $\mu$m (red curve). (**C**) An approximate phase diagram of protein-membrane binding. The dashed lines delineate regimes, as explained further in Appendix 1.

DOI: https://doi.org/10.7554/eLife.40472.004

The following figure supplement is available for figure 2:

**Figure supplement 1.** Energetic modeling of MreB binding and deformation of vesicles.

DOI: https://doi.org/10.7554/eLife.40472.005

grossly deforms the membrane for small values of $p$, the preferred binding orientation is robust to changes in $p$ (**Figure 2B**). In fact, across a wide range of parameter values including the filament bending rigidity ($B = \pi Y r_f^4/4$), the filament intrinsic curvature ($\kappa_s$), and the membrane pressure difference ($p$), a preferred binding orientation exists and coincides with the direction of largest principal curvature for any membrane which is less curved than the filament. In the case of $p = 0$, as discussed in Appendix 1, the prediction that MreB binding induces large membrane deformations is consistent with cryo-electron microscopy images of MreB binding to vesicles (**Figure 1A**) (**Salje et al., 2011**; **van den Ent et al., 2014**). Importantly, the energetic penalties for deviatory binding conformations are larger than the energy of thermal fluctuations across a large range of $p$, suggesting the empirically observed variation in binding orientation (**Figure 1C**) to be caused by other sources of stochasticity. The energetic penalties are also decreased in wider membranes for both small and physiological values of $p$, the latter of which is consistent with the gradual widening of the distributions of MreB trajectory angles in wider *B. subtilis* protoplasts (**Hussain et al., 2018**).

In general, the mechanics of filament binding are well described by the pressure difference across the membrane and the filament bending energy, which can be viewed as order parameters that largely dictate whether the filament predominantly bends the membrane, bends to conform to the membrane, or both. An approximate phase diagram for MreB binding to any membrane which is less curved than the filament can be determined (**Figure 2C** and Appendix 1). Below, we suppose the membrane surface to be less curved than the filament—so that it is always energetically favorable to orient along directions of largest membrane curvature—and model filament translocation along these directions.

## Dynamics of translocation

We next examine how the translocation of MreB filaments, once bound to the membrane, affects their distribution in different geometries. Inside cells, MreB filaments move along the membrane in the direction of their orientation (**Hussain et al., 2018**; **Olshausen et al., 2013**) (**Figure 1C**). This directional and processive motion is driven by cell wall synthesis (**Sliusarenko et al., 2010**; **van Teeffelen et al., 2011**; **Olshausen et al., 2013**), and filaments may reorient according to different membrane geometries (**Hussain et al., 2018**). Thus, highly-bent MreB filaments translocate along the direction of largest curvature, a direction that minimizes the energetic cost of binding. As discussed below, this hypothesis is supported by observations that (1) MreB moves circumferentially in live *B. subtilis* cells, but this motion becomes disoriented if cells become round, (2) circumferential motion is re-established when round cells are confined into rods, (3) MreB moves directionally in bulges protruding from round cells, and (4) MreB filaments rapidly translocate out of poles in rods, reorienting when filaments reach the cylindrical bulks (**Hussain et al., 2018**). Assuming translocation on a static surface, we may model the trajectories of filaments as biased random walks as follows (with more details provided in Appendix 1). The case of a dynamical surface, as expected for MreB-directed growth, can be considered similarly. Note that a 'biased random walk' refers to a succession of random steps which may be processive: while the mean-squared displacement of a filament will be approximately quadratic, and not linear, in time, the processive motion we consider is random only because the translocation direction can deviate from directions of largest membrane curvature due to sources of stochasticity (**Figure 1C**). We will show that our biased random walk model of filament trajectories leads to predictions of MreB localization.

We consider the membrane as a parametric surface, $\mathbf{r} = r(u,v)$, embedded in three-dimensional space ($\mathbb{R}^3$) with surface coordinates $u$ and $v$ and a filament as a point on this surface which, at any moment in time, translocates along the *largest principal direction* $\mathbf{d}$—that is, the direction of largest curvature of the surface. As $\mathbf{d}$ is a vector in $\mathbb{R}^3$, arbitrarily moving in the direction of $\mathbf{d}$ may move the filament off of the surface. To define the translocation consistently, we set $\eta = \cos^{-1}\frac{\mathbf{d}\cdot\mathbf{r}_\theta}{\|\mathbf{d}\|\cdot\|\mathbf{r}_\theta\|}$, where $\eta$ is an angular deviation from the largest principal direction on the surface introduced by possible sources of stochasticity, the modified direction corresponds to an angle $\theta$ relative to the $u$-axis in parametric coordinates, $\mathbf{r}_\theta \in \mathbb{R}^3$ is the derivative of $\mathbf{r}$ in the direction of $\theta$, and distances are defined by the surface metric. Translocating along an angle $\theta$ with respect to the $u$-axis in $(u,v)$-coordinates then ensures that the filament remains on the surface, and the direction of translocation corresponds to that on a patch of $\mathbf{r}$.

As a discrete-time flow in $(u, v)$-coordinates, and with suitable units of time so that the filament may reorient at every timestep, the 2D equation of filament motion is

$$X_{n+1} = X_n + \chi_n \ell_n (\cos \theta_n, \sin \theta_n), \tag{3}$$

where $X_n$, $\ell_n$, and $\theta_n$ are the position, step size, and translocation angle, respectively, of the filament at a timestep $n$. Here $\theta_n$ is the value of $\theta$ computed at the surface point corresponding to $X_n$ and assuming $\eta \sim \mathcal{N}(0, \sigma^2)$—that is, the angular noise is normally-distributed, with mean zero and a variance, $\sigma^2$, to be inferred from data—and note that the *translocation noise*, $\sigma$, may depend on quantities such as the principal curvatures, as discussed later. $\chi_n$ is a random sign, which accounts for the possibility of both left-handed and right-handed translocation, and may not substantially vary in $n$ if the filament does not backtrack, as is assumed for the remainder of this work. We assume that $\ell_n$ satisfies an integral equation which relates it to a constant and finite filament step size, $L$, on the surface (Appendix 1), and note that inertia in MreB motion, as measured previously by the velocity autocorrelation (*Kim et al., 2006*; *van Teeffelen et al., 2011*), is modeled by the finite step size. Note that, when both principal curvatures are equal, so that $\mathbf{d}$ is not well defined, we will assume steps of uniformly random angles and size $L$ on the surface. Furthermore, for finite step sizes, *Equation (3)* extrapolates the translocation direction in surface coordinates: while motion along the largest principal direction can be maintained irrespective of parameterization by parallel transport, we anticipate the distinction from *Equation (3)* to be insignificant for many geometries due to the small step sizes considered in this work. Determining the probability distribution of $X$ then determines the probability distribution of the filament on $\mathbf{r}$, and this can be done analytically and numerically for several geometries as discussed below.

We next consider the dynamics of activating and deactivating filaments as follows. We suppose that a filament may be activated at a position $X$ at any timestep at a constant rate $k \geq 0$ with probability proportional to the membrane surface area, $dA(X)$, and deactivated at a constant rate $\lambda \geq 0$ which determines the filament's processivity—that is, the mean number of steps that a filament takes on the membrane surface before becoming inactive (*Figure 3A*). The case of $k$ being dependent on fields, such as mechanical strains (*Wong et al., 2017*), can be considered similarly but is not necessary for the results below. An ensemble of filaments produced by such dynamics will exhibit filament numbers, $N_F$, that vary in space and time, and likewise for the filament concentration $C_F(X, n) = N_F(X, n)/dA(X)$. Below, we discuss characteristic parameter values relevant to MreB and show that the dynamics of *Equation (3)* gives rise to localization. We then examine the model in detail and describe how localization depends on different parameters of the model.

## Implications to MreB localization

Previous fluorescence microscopy measurements provide estimates for the step size ($L$), deactivation rate ($\lambda$), and translocation noise ($\sigma$) of MreB filaments in cells. We assume $L$ to be $200\,\mathrm{nm}$ as a modeling choice, but show in Appendix 1 that the results discussed below are qualitatively similar for significantly larger $L$ ($\sim 2\,\mu\mathrm{m}$). Similar experiments have estimated the persistence time of MreB filaments in *E. coli* as ~5 min (*Ursell et al., 2014*), while a characteristic translocation noise of $\sigma \approx 0.3\,\mathrm{rad}$ in *B. subtilis* has been found separately by (1) measuring filament trajectory angles relative to the midline and (2) measuring binding angles in confined protoplasts (*Figure 1C*) (*Hussain et al., 2018*). While we assume the values of $\lambda$ and $\sigma$ to be based on these measurements, we examine the effects of varying $\lambda$ and $\sigma$ in the following section. Furthermore, in recent studies, rod-shaped cells have been perturbed to be in geometries other than a spherocylinder (*Ursell et al., 2014*; *Wong et al., 2017*; *Hussain et al., 2018*; *Renner et al., 2013*; *Amir et al., 2014*). As the distribution of MreB filament angles gradually becomes broader as *B. subtilis* cells become wider (*Hussain et al., 2018*), it may also be reasonable to suppose that $\sigma$ depends on the difference, $\Delta c$, of principal curvatures at the location of any MreB filament: $\sigma = \alpha (\Delta c)^{-1}$, where $\alpha \approx 0.6\,\mathrm{rad} \cdot \mu\mathrm{m}^{-1}$ is a constant of proportionality determined by experimental data. While all our results pertaining to MreB below assume this dependence as to be consistent with data, we show in Appendix 1 that our results are similar for different dependencies of $\sigma$ on $\Delta c$.

Given the aforementioned parameters, *Equation (3)* leads to predictions for the statistics of the filament position ($X$) and the ensuing filament concentration ($C_F$) across different membrane geometries. For a cylindrical cell, analytical expressions for the statistics of $X$ show that translocation noise

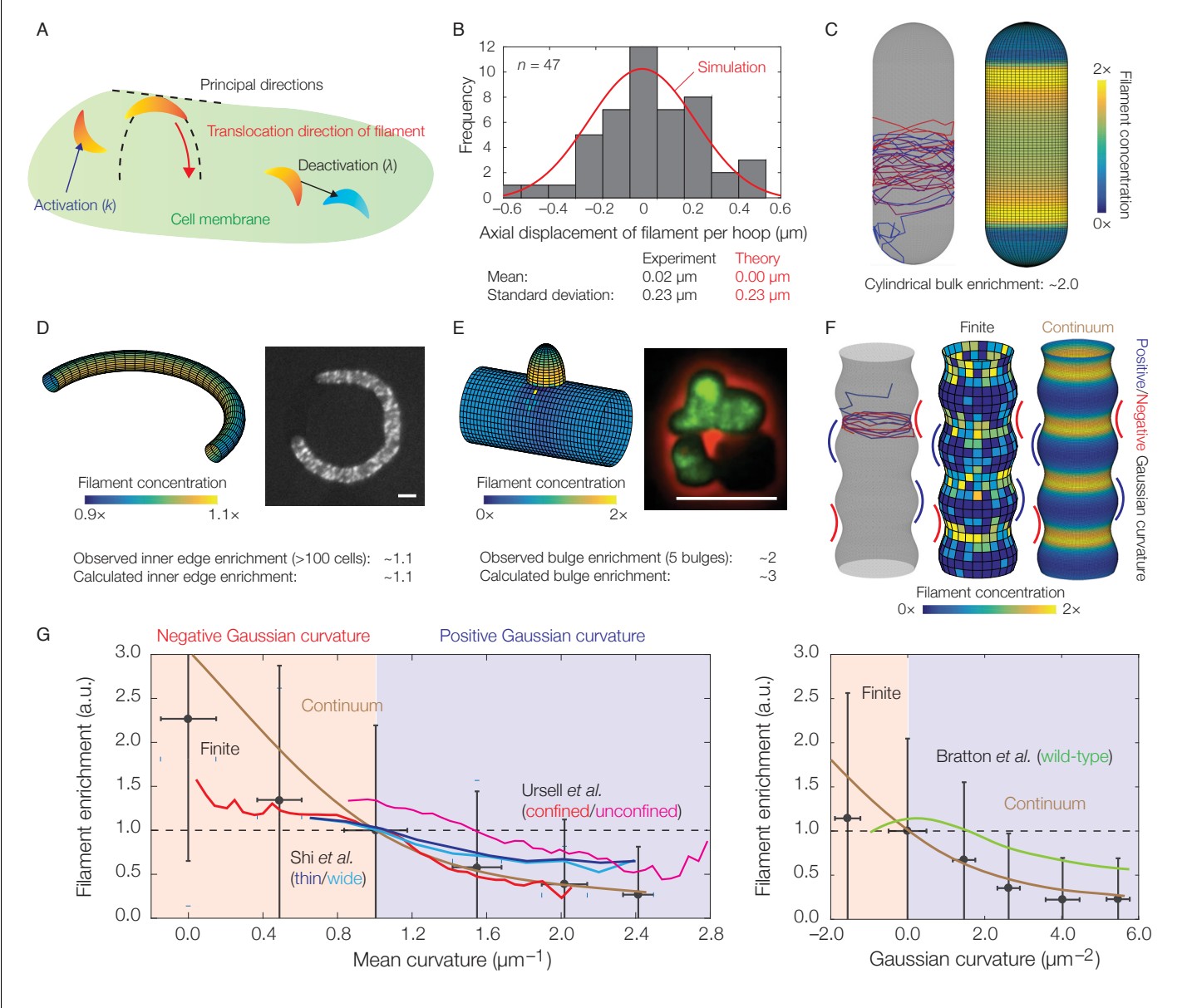

**Figure 3.** Dynamics of translocation and implications to MreB localization. In this figure, the localization of MreB filaments, which exhibit a finite processivity and are assumed to follow the parameter values summarized in *Supplementary file 2*, is shown. (A) A schematic of the model of filament translocation. The filament is modeled as a point which moves processively in a direction determined by the principal curvatures. (B) Histogram of the axial displacements per hoop translocated of 47 MreB filaments in *B. subtilis* cells (*Hussain et al., 2018*), along with the theoretical prediction and a simulation result shown. (C) Langevin simulation (left) of *Equation (3)* and numerical result (right) for the filament concentration, $C_F$, on a spherocylinder, for parameter values relevant to *E. coli*. Here and below, blue and red denote starts and ends of trajectories, respectively, details of simulations and numerics are provided in Appendix 1, and $C_F$ is found by solving the Fokker-Planck equation corresponding to *Equation (3)*. (D) (Left) Numerical result for $C_F$ on a bent rod, for parameter values relevant to *E. coli*. (Right) Representative fluorescence microscopy image of an *E. coli* cell confined to a donut-shaped microchamber, with MreB tagged by green fluorescent protein (GFP), from *Wong et al. (2017)*. The inner edge enrichment is calculated as described in *Wong et al. (2017)*, and the scale bar indicates 1 μm. (E) (Left) Numerical result for $C_F$ on a cylinder with a bulge, for parameter values relevant to *B. subtilis*. (Right) Representative fluorescence microscopy image of a deformed *B. subtilis* cell with a bulge and GFP-tagged MreB, from *Hussain et al. (2018)*. The bulge enrichment is calculated as a ratio of average pixel intensities, and the scale bar indicates 5 μm. (F) Langevin simulation and numerical results for $C_F$ on a cylinder with surface undulations in both the finite (Langevin, total of ~500 filaments) and continuum (Fokker-Planck) cases, for parameter values relevant to *E. coli*. (G) A plot of (left) the mean curvature and (right) the Gaussian curvature against filament enrichment for the figures shown in (F) (Langevin simulation, black; continuum case, brown), with empirically observed relations from confined and unconfined MreB-labeled *E. coli* cells (red and magenta) from *Ursell et al. (2014)*, thin and wide mutant *E. coli* cells (blue and cyan) from *Shi et al. (2017)*, and wild-type *E. coli* cells (green) from *Bratton et al. (2018)* overlaid. Error bars denote one standard deviation in the Langevin

*Figure 3 continued on next page*

**eLIFE** Research advance

Microbiology and Infectious Disease | Physics of Living Systems

*Figure 3 continued*

simulation, and 1 a.u. equals the mean of $C_F$ when the mean curvature is 1 µm$^{-1}$ (left) and when the Gaussian curvature is 0 µm$^{-2}$ (right). Note that the magenta and green curves are not normalized according to this convention.

DOI: https://doi.org/10.7554/eLife.40472.006

The following figure supplements are available for figure 3:

**Figure supplement 1.** Curvature-based translocation on cylinders with bulges.

DOI: https://doi.org/10.7554/eLife.40472.007

**Figure supplement 2.** Correlations between Gaussian and mean curvatures for, and translocation directions in, cylinders with undulations of different wavelengths.

DOI: https://doi.org/10.7554/eLife.40472.008

**Figure supplement 3.** Effects of curvature-dependent translocation noise and varying filament properties on model predictions.

DOI: https://doi.org/10.7554/eLife.40472.009

does not significantly affect the mean or variance of the circumferential displacement of a filament (Appendix 1). In contrast, the value of $\sigma \approx 0.3\,\mathrm{rad}$ corresponds to a standard deviation of approximately $0.2\,\mu\mathrm{m}$ for the axial displacement of a filament per hoop of wall material inserted. This value is consistent with experimental measurements (*Figure 3B*), showing that deviations from a circumferential translocation direction can significantly contribute to wall insertions in the axial direction and disordered wall architecture.

MreB filaments have been observed to be depleted from the hemispherical poles of spherocylindrical cells compared to the cylindrical bulks (*Kawazura et al., 2017*; *Ursell et al., 2014*). Observations of filament dynamics revealed a possible explanation: MreB filaments reorient rapidly in, and translocate out of, the poles and into the bulks, where motion then becomes aligned (*Hussain et al., 2018*). Consistent with this observation, simulations of *Equation (3)* on a spherocylindrical surface show that the concentration of filaments in the bulk is enhanced (*Figure 3C*). The average filament concentration is predicted to be approximately two-fold higher in the bulk than the poles, in agreement with experimental measurements in *E. coli* (*Ursell et al., 2014*). Simulations of *Equation (3)* on a toroidal surface are also quantitatively consistent with prior measurements of MreB fluorescence in *E. coli* cells confined to donut-shaped microchambers, which have shown that MreB intensity is increased at the inner edges by a factor of ~1.1 relative to the midlines (*Figure 3D*) (*Wong et al., 2017*). For a spherocylinder, filament enrichment arises because the cylindrical bulk retains filaments: oriented motion is preserved in the bulk, while disordered motion at the poles eventually becomes ordered. In contrast, filament enrichment arises in a curved cell because filaments become uniformly distributed along circumferential hoops. The smaller arclength along the inner edge then results in a greater density of filaments.

In our previous study, we found that MreB rotation and localization at small protrusions in *B. subtilis* protoplasts preceded rod shape generation from these protrusions (*Hussain et al., 2018*). To model the geometry observed in these experiments, we consider a cylindrical body with a protruding bulge in which filament trajectories become parallel to the cylinder long axis. Simulations of *Equation (3)* on this geometry reveal that the filament concentration is larger in the bulge and that the predicted enrichment is quantitatively consistent with the MreB enhancement observed in bulged cells, without any fitting parameters (*Figure 3E* and *Figure 3—figure supplement 1*). Similar to the case of a spherocylinder, localization arises due to the bulge attracting filaments. The dynamics of *Equation (3)* therefore results in localization which contributes to de novo generation of rod shape.

Finally, previous work has examined MreB localization in *E. coli* cells (1) with submicron-scale shape fluctuations or (2) confined in sinusoidal chambers (*Ursell et al., 2014*; *Shi et al., 2017*; *Bratton et al., 2018*). The empirically observed magnitudes of MreB enrichment at regions of negative Gaussian curvature or small mean curvature in these studies are consistent with our modeling. To model the cell shapes observed in these experiments, we consider filament translocation on a geometry with both negative and positive Gaussian curvatures and undulations of smaller wavelengths than the surface size (*Figure 3F* and *Figure 3—figure supplement 2*). As discussed in Appendix 1, the Gaussian and mean curvatures in this geometry are positively correlated and consistent with experimental observations (*Ursell et al., 2014*). For this geometry, filament translocation results in increased values of concentration ($C_F$) at regions of negative Gaussian curvature or small

mean curvature (*Figure 3F*). This effect arises because the principal curvatures away from these regions reorient filaments axially, instead of circumferentially, so that regions of negative Gaussian curvature or small mean curvature attract filaments. Furthermore, the magnitude of this enhancement is consistent with the amount of MreB enrichment observed (*Figure 3G* and *Figure 3—figure supplement 3*), demonstrating that translocation dynamics alone can negatively correlate filament concentration with Gaussian or mean curvature in cells with similar short wavelength undulations.

## Dependence of localization on processivity and Gaussian curvature

As we anticipate our model to be applicable to general filament systems, we now explore the response of the filament concentration ($C_F$) to (1) different parameter values and (2) other geometries. We fix the filament step size ($L$) and suppose the translocation noise ($\sigma$) and the deactivation rate ($\lambda$) to be constants which are varied within a broad range. We show in Appendix 1 that, for any value of processivity and zero translocation noise, $C_F$ is uniform over the surface of an ellipsoid, as is generally the case for any surface when the processivity is small (or, equivalently, $\lambda$ is large). In contrast, in the case of small $\lambda$ corresponding to large processivity—a limiting case that is relevant to MreB—and over a range of $\sigma$, $C_F$ is larger at the inner edge of a torus, at the inner edge of a helix, and at the tips of an ellipsoid (*Figure 4A* and *Figure 4—figure supplements 1* and *2*). As discussed above, localization occurs geometrically in these cases due to the filament number ($N_F$) becoming uniform over the surface and spatial variations in the surface area element. The magnitude of the localization can be quantitatively predicted by geometric parameters alone (Appendix 1). The mechanism underlying localization is different for a spherocylinder or a bulged cylinder, for which surface regions attract filaments. Nevertheless, a nonzero processivity is required for localization even in geometries which attract filaments (*Figure 4B* and *Figure 4—figure supplements 1* and *2*).

Since filament enrichment depends on both processivity and geometry, we wondered if the localization of processive filaments always correlates with the Gaussian or mean curvatures, regardless of overall geometry. Although *Figure 3G* demonstrates that filament enrichment correlates with negative Gaussian curvature or small mean curvature in a specific, undulating geometry, this correlation is reversed in bulged cylinders (*Figure 3E*). Furthermore, *Figure 4C* illustrates a surface of zero Gaussian curvature exhibiting regions which attract filaments, as filaments change from moving circumferentially to moving axially in such regions (see *Figure 4—figure supplement 3* for additional details). Examining *Equation (3)* on different surfaces therefore shows that $C_F$ need not depend on Gaussian curvature at all, and the dynamics modeled in this work cannot act as a generic mechanism for sensing Gaussian curvature.

Finally, while large filament bundles or twist have not been observed in MreB filaments reconstituted in vitro (*Salje et al., 2011*; *van den Ent et al., 2014*; *Hussain et al., 2018*), it is possible that general filament systems could exhibit these properties (*Wang and Wingreen, 2013*; *Quint et al., 2016*). The binding and activation of twisted filaments may also depend on membrane Gaussian curvature, as previously demonstrated (*Quint et al., 2016*). We systematically explore the effects of varying filament bending rigidity, filament twist, and Gaussian curvature-dependent activation in Appendix 1, where we show that our model predictions remain largely robust across a broad range of these parameters (*Figure 4—figure supplement 4*). Thus, we expect filament dynamics to contribute to localization in different filament systems, regardless of the details of filament rigidity, twist, and other parameters of our model.

## Discussion

An outstanding problem in bacterial physiology has been to understand how short and disconnected filaments distribute themselves within cells to conduct different cellular functions (*Eun et al., 2015*). In this work, we have examined an aspect of this problem by modeling the direct binding of protein filaments to membranes and the curvature-based translocation of an ensemble of such filaments. Our results provide a theoretical framework for prior work examining MreB dynamics and localization (*Hussain et al., 2018*; *Salje et al., 2011*; *Wong et al., 2017*; *Ursell et al., 2014*; *Shi et al., 2017*; *Bratton et al., 2018*; *Renner et al., 2013*). Furthermore, our results are consistent with the cellular localization observed in all these works and demonstrate that filament motion alone can correlate enrichment with Gaussian curvature in specific geometries. Our work may be extended by modeling an evolving membrane surface, as expected for MreB-directed growth, and it would be

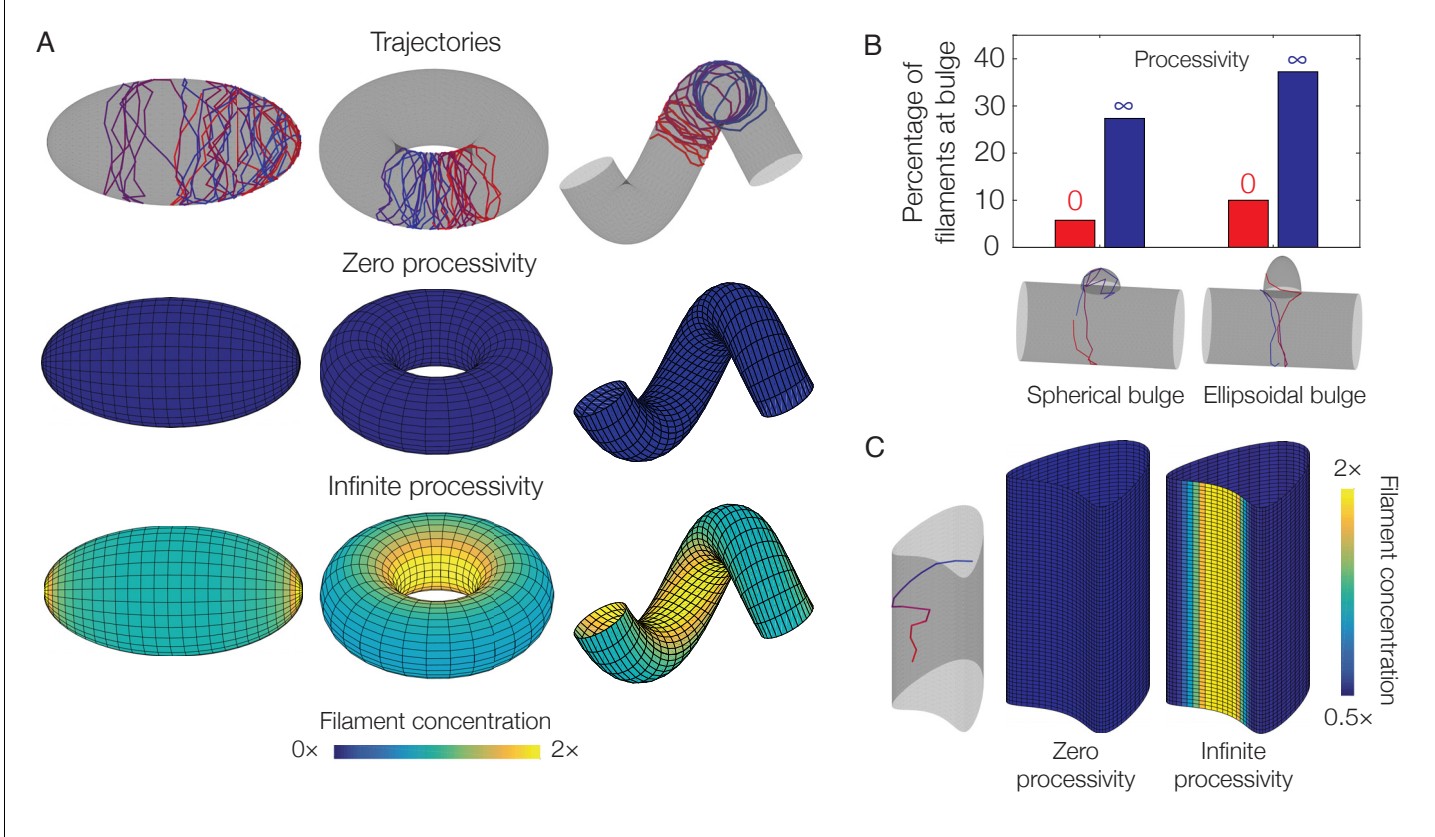

**Figure 4.** Dependence of localization on processivity and Gaussian curvature. (**A**) Langevin simulations of *Equation (3)* and numerical results for $C_F$, the filament concentration, on different surfaces. Note that cases of zero processivity correspond to uniform distributions and that we have considered the limiting cases of zero and infinite processivity, along with a constant value of the translocation noise ($\sigma$), here. *Figure 3* shows numerical results for the case of a large, but finite, processivity and a principal curvature-dependent translocation noise relevant to MreB. (**B**) Plot of the percentage of total filaments contained in a bulge, for the two different geometries indicated in the limits of zero and infinite processivity. (**C**) Langevin simulation and numerical result for $C_F$ on a non-circular cylinder in the limits of zero and infinite processivity.

DOI: https://doi.org/10.7554/eLife.40472.010

The following figure supplements are available for figure 4:

**Figure supplement 1.** Curvature-based translocation on a torus and a helix.
DOI: https://doi.org/10.7554/eLife.40472.011

**Figure supplement 2.** Curvature-based translocation on an ellipsoid.
DOI: https://doi.org/10.7554/eLife.40472.012

**Figure supplement 3.** Curvature-based translocation on a geometry with zero Gaussian curvature.
DOI: https://doi.org/10.7554/eLife.40472.013

**Figure supplement 4.** Effects of filament twist, flexural rigidity, and Gaussian curvature-dependent activation on model predictions.
DOI: https://doi.org/10.7554/eLife.40472.014

intriguing to explore whether and how principal curvature-based translocation contributes to determining cell width.

The main contribution of this work is to show that the biological results of MreB localization, as observed in many different experiments involving a range of cell shapes (*Hussain et al., 2018*; *Wong et al., 2017*; *Ursell et al., 2014*; *Shi et al., 2017*; *Bratton et al., 2018*), can arise from processivity and principal curvature-dependent motion alone. Our study therefore helps to unravel how rod shape formation may be achieved through subcellular-scale mechanisms (*Amir and van Teeffelen, 2014*; *Shi et al., 2018*; *Surovtsev and Jacobs-Wagner, 2018*). More broadly, our work shows that the localization of translocating protein filaments can vary significantly depending on membrane geometry. This paves the way for exploring similar behavior in other contexts, such as bacterial cytokinesis and eukaryotic membrane trafficking and transport. For example, in bacterial cytokinesis,

filaments of the tubulin homolog FtsZ assemble at, and treadmill around, the septum, a process which directs the insertion of new PG and constricts the cell (*Bisson-Filho et al., 2017*; *Yang et al., 2017*). Like MreB, FtsZ filaments are curved and could orient along the largest principal direction on membranes through bending alone (*Osawa et al., 2009*; *Erickson et al., 2010*). Treadmilling along such directions would then allow filaments to drive PG synthesis circumferentially at the septum.

Aside from MreB and FtsZ, septins, BAR-domain-containing proteins, dynamins, and endoproteins are known to exhibit similar, curvature-dependent membrane binding behaviors important for membrane trafficking, growth, and movement in both prokaryotes and eukaryotes (*Baumgart et al., 2011*; *Zimmerberg and Kozlov, 2006*; *McMahon and Gallop, 2005*; *Peter et al., 2004*; *Low and Löwe, 2006*; *Raiborg and Stenmark, 2009*; *Teo et al., 2006*; *Kostelansky et al., 2007*). Like MreB filaments, many such proteins sense membrane curvature through mechanical deformations of either the membrane or the protein itself. Unlike MreB or FtsZ, these proteins do not translocate; rather, they often induce membrane curvature to facilitate downstream processes. One example is BAR-domain-containing proteins, which scaffold higher-order assemblies of dynamin that actively constrict for vesicle scission (*McMahon and Gallop, 2005*). It would be interesting to apply the methods introduced here to this and other biological systems where molecules are known to bind to membranes or sense membrane curvature. These systems are widespread and involved in pathogenesis (*Baumgart et al., 2011*; *Frost et al., 2009*), cell division (*Renner et al., 2013*; *Ramamurthi and Losick, 2009*; *Ramamurthi et al., 2009*; *Frost et al., 2009*), intracellular trafficking (*Zimmerberg and Kozlov, 2006*; *McMahon and Gallop, 2005*; *Raiborg and Stenmark, 2009*; *Ford et al., 2002*; *Frost et al., 2009*; *Römer et al., 2007*), and cell migration (*Frost et al., 2009*; *Zhao et al., 2013*). The mathematical model introduced in this work, which requires minimal assumptions as to how filaments bind to and translocate on membranes, should be widely applicable to these and other broader contexts.

## Acknowledgements

FW was supported by the National Science Foundation Graduate Research Fellowship under grant no. DGE1144152 and the Quantitative Biology Initiative at Harvard. ECG was supported by the National Institutes of Health under grant no. DP2AI117923-01, the Smith Family Award, and the Searle Scholar Fellowship. AA was supported by the Materials Research and Engineering Center at Harvard, the Kavli Institute for Bionano Science and Technology at Harvard, and the Alfred P Sloan Foundation. ECG and AA were supported by the Volkswagen Foundation. We thank Carl Wivagg, Saman Hussain, Ned Wingreen, and Siyuan (Steven) Wang for discussions and Sven van Teeffelen, Jie Lin, and Po-Yi Ho for comments on the manuscript.

## Additional information

### Funding

| Funder | Grant reference number | Author |
| --- | --- | --- |
| National Science Foundation | DGE1144152 | Felix Wong |
| Quantitative Biology Initiative at Harvard | | Felix Wong |
| National Institutes of Health | DP2AI117923-01 | Ethan C Garner |
| Smith Family Award | | Ethan C Garner |
| Searle Scholar Fellowship | | Ethan C Garner |
| Volkswagen Foundation | | Ethan C Garner Ariel Amir |
| Materials Research and Engineering Center at Harvard | | Ariel Amir |
| Kavli Institute for Bionano Science and Technology at Harvard | | Ariel Amir |

Alfred P. Sloan Foundation                                    Ariel Amir

The funders had no role in study design, data collection and interpretation, or the decision to submit the work for publication.

#### Author contributions
Felix Wong, Conceptualization, Software, Formal analysis, Validation, Investigation, Visualization, Methodology, Writing—original draft, Writing—review and editing; Ethan C Garner, Conceptualization, Formal analysis, Funding acquisition, Investigation, Methodology, Writing—original draft, Writing—review and editing; Ariel Amir, Conceptualization, Formal analysis, Supervision, Funding acquisition, Investigation, Methodology, Writing—original draft, Writing—review and editing

#### Author ORCIDs
Felix Wong [iD] http://orcid.org/0000-0002-2309-8835
Ethan C Garner [iD] https://orcid.org/0000-0003-0141-3555
Ariel Amir [iD] http://orcid.org/0000-0003-2611-0139

#### Decision letter and Author response
Decision letter https://doi.org/10.7554/eLife.40472.023
Author response https://doi.org/10.7554/eLife.40472.024

## Additional files

#### Supplementary files
• Supplementary file 1. Variables used, or calculated, in the model of filament binding and their numerical values.
DOI: https://doi.org/10.7554/eLife.40472.015

• Supplementary file 2. Variables used, or calculated, in the model of filament translocation and their numerical values for *E. coli*.
DOI: https://doi.org/10.7554/eLife.40472.016

• Transparent reporting form
DOI: https://doi.org/10.7554/eLife.40472.017

#### Data availability
All data generated or analyzed during this study are included in the manuscript and supporting files.

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

## Appendix 1

DOI: https://doi.org/10.7554/eLife.40472.018

# 1. Mechanics of binding

## 1.1. Model of a protein filament binding to a membrane

We consider the protein as a filament with monomeric subunits that bind to a membrane in an energetically favorable manner, such as burial of hydrophobic residues (**Hussain et al., 2018**). When a filament binds to a membrane, an energetic cost $E_{\mathrm{def}}(\ell_b)$ is associated to deformations which deviate from a position at mechanical equilibrium, while the free energy may be lowered by an amount $E_{\mathrm{int}}(\ell_b)$ due to interaction (**Figure 2—figure supplement 1a**). Both the deformation and interaction energies are expressed as functions of the bound filament length, $\ell_b$, which is less than or equal to the total filament length, $L_f$. We wish to minimize the free energy due to filament binding, $\Delta E = E_{\mathrm{def}} - E_{\mathrm{int}}$. If $\Delta E(\ell_b)$ is negative, then it is energetically favorable for the filament to bind to the membrane along a length $\ell_b$. We estimate $E_{\mathrm{int}}$ as

$$E_{\mathrm{int}}(\ell_b) = \varepsilon_{\mathrm{int}}\ell_b, \quad \varepsilon_{\mathrm{int}} \equiv N_{\mathrm{int}}\varepsilon_0/L_f, \tag{S1}$$

where $N_{\mathrm{int}}$ denotes the total number of membrane binding sites of the filament and $\varepsilon_0$ denotes an independent and additive single binding site energy, which is given for MreB along with other parameter values in **Supplementary file 1**.

We assume that the binding sites are arranged linearly along the filament, and in particular, that the filament is not twisted. In this case, it suffices to account only for filament bending, and we may decompose the deformation energy $E_{\mathrm{def}}$ into the bending energy of the filament, $E_{\mathrm{bend}}$, and the deformation energy of the membrane, $E_{\mathrm{mem}} : E_{\mathrm{def}} = E_{\mathrm{bend}} + E_{\mathrm{mem}}$. With notation as in the main text, we model the filament as a curved, cylindrical elastic rod with a circular cross-section of radius $r_f$ and curvature $1/R_s$, so that the elastic energy density per unit length of bending the filament from a curvature of $1/R_s$ to a curvature of $1/R$ is

$$\varepsilon_{\mathrm{bend}} = \frac{\pi Y r^4}{8}\left(\frac{1}{R} - \frac{1}{R_s}\right)^2 = \frac{B}{2}\left(\frac{1}{R} - \frac{1}{R_s}\right)^2, \tag{S2}$$

where $Y$ is the elastic modulus of the filament and $B = \pi Y r_f^4/4$ is its flexural rigidity (**Landau and Lifshitz, 1970**). The resulting filament bending energy is $E_{\mathrm{bend}} = \varepsilon_{\mathrm{bend}}\ell_b$. For simplicity, we have assumed the filament to be bent uniformly, but the case of a curvature which varies with position along the filament length can be considered similarly.

As stated in the main text, we assume an isotropic, fluid bilayer membrane, where there is no in-plane shear modulus and the only in-plane deformations are compressions and expansions. The membrane free energy assumes the form of **Equation (1)** in the main text. The mechanical energy required to bend a membrane from a surface $S_0$, with a mean curvature $H_0$, to a surface $S$ is then the difference of the corresponding free energies:

$$E_{\mathrm{mem}} = \min_S\left[2k_b\int_S(H^2 - H_0^2)dA - p\int dV\right], \tag{S3}$$

where the surface integrals of the Gaussian curvature are topological invariants by the Gauss-Bonnet theorem and therefore cancel in the difference, and the volume integral is understood to be a difference of the volumes in the deformed and undeformed states.

Minimizing $\Delta E$ requires the minimization of $E_{\mathrm{mem}}$ given some value $1/R$ of the deformed filament curvature. To minimize $E_{\mathrm{mem}}$ in **Equation (S3)**, we assume that the surface $S$ can be parameterized in the Monge gauge $h = h(x, y)$, where $(x, y) \in \mathbb{R}^2$, and furthermore that $|\nabla h| \ll 1$: this means that the membrane surface is not excessively curved or kinked. We assume the same for the undeformed surface $S_0$, which is parameterized by a function $h_0$ in the Monge gauge. In the case of binding to a cylindrical membrane, we may, for instance, take the undeformed surface to correspond to a cylinder with radius $R_{\mathrm{cell}}$:

$$h_0(x,y) = R_{\text{cell}} - \sqrt{R_{\text{cell}}^2 - x^2}, \quad |x| < R_{\text{cell}}. \tag{S4}$$

In the Monge gauge, the mean curvature can be expanded as $H = \frac{1}{2}\nabla^2 h + O[(\nabla h)^2]$, where the big-$O$ notation signifies $|H - \frac{1}{2}\nabla^2 h| \leq M(\nabla h)^2$ when $0 < (\nabla h)^2 < \delta$ for some positive numbers $\delta$ and $M$. The membrane bending energy in **Equation (S3)** can then be rewritten with the Laplacian, $\Delta$, as

$$E_{\text{mem}} = \min_h \mathcal{F}[h], \quad \mathcal{F}[h] = \frac{k_b}{2}\int_\Omega \left[(\Delta h)^2 - (\Delta h_0)^2\right] dxdy - p\int_\Omega (h - h_0)dxdy, \tag{S5}$$

for some domain $\Omega \subset \mathbb{R}^2$ of $h$ and $h_0$ not containing the domain $U$ of the filament surface (**Figure 2—figure supplement 1a**). Setting the first variation of $\mathcal{F}[h]$ to zero, we find that the equilibrium membrane shape is given by the solution of the shape equation

$$\Delta^2 h = \frac{p}{k_b}, \tag{S6}$$

where $\Delta^2$ is the biharmonic operator. **Equation (S6)** is subject to the Dirichlet boundary conditions

$$\begin{cases} h(x,y) = \phi(x,y) & (x,y) \in \partial\Omega \\ \Delta h(x,y) = \psi(x,y) & (x,y) \in \partial\Omega. \end{cases} \tag{S7}$$

Here $\phi$ and $\psi$ are indicator functions defined by their values on the boundary of $\Omega$, $\partial\Omega$. In the case of a cylindrical membrane, for instance,

$$\begin{cases} \phi(x,y) = h_0(x,y) & (x,y) \in \partial\Omega - \partial U \\ \phi(x,y) = p_0(x,y) & (x,y) \in \partial U, \end{cases} \quad \begin{cases} \psi(x,y) = 1/R_{\text{cell}} & (x,y) \in \partial\Omega - \partial U \\ \psi(x,y) = 2C_0 & (x,y) \in \partial U, \end{cases} \tag{S8}$$

where, as above, $h_0$ parameterizes the undeformed surface $S_0$, $p_0(y)$ is a quadratic function describing the values of the filament height at the curve $\partial U$ parameterizing the binding region, and $C_0$ is the mean curvature of the filament along $\partial U$. Thus, the first condition in **Equation (S7)** comes from imposing continuity of membrane height with respect to the filament surface, while the second condition comes from imposing continuity of mean curvature. In live cells, we treat the periplasm as a rigid, undeformable body (**Hussain et al., 2018**), so that, for instance, $p_0(y) \geq h_0(x,y)$ for all $(x,y) \in \Omega$ when $h_0$ assumes the form of **Equation (S4)** above. With the boundary conditions of **Equation (S7)**, **Equation (S6)** can be conveniently decoupled as two Poisson equations, each with Dirichlet boundary conditions:

$$\begin{cases} \Delta h(x,y) = f(x,y) & (x,y) \in \Omega \\ h(x,y) = \phi(x,y) & (x,y) \in \partial\Omega, \end{cases} \quad \begin{cases} \Delta f(x,y) = p/k_b & (x,y) \in \Omega \\ f(x,y) = \psi(x,y) & (x,y) \in \partial\Omega. \end{cases} \tag{S9}$$

Since any solution to the Poisson equation with Dirichlet boundary conditions is unique, the decomposition above yields a unique solution for $h$.

As the foregoing considerations assume a fixed $\Omega$ in determining h, the size of $\Omega$ is an additional variable that must be considered. Since shape space is infinite-dimensional, determining $h$ for an arbitrarily large $\Omega$ does not necessarily imply that the global minimum of $\Delta E$ is achieved; neither does it necessarily determine the appropriate decay length of the indentation, since $h(x,y) = h_0(x,y)$ is generally not a solution to **Equation (S9)**. It is possible that $\Delta E$ may be minimized at a finite $\Omega$. Due to the boundary conditions (**Equation (S7) and (S8)**), solutions of **Equation (S9)** over finite $\Omega$ are continuous, with continuous mean curvatures, and could in fact be physically plausible. This subtlety can be addressed by choosing $\Omega$ so that the numerically computed value of $\Delta E$ is minimal among differently sized $\Omega$, as discussed below.

## 1.2. Finite element solutions of the shape equation

Given values of $k_b, p, R_{\text{cell}}, C_0$, and the filament height along $\partial U$, we numerically solved *Equation (S6)* by individually solving the decoupled equations, *Equation (S9)*, with a two-dimensional finite element Poisson equation solver (*Figure 2—figure supplement 1b–c*) (*Burkardt, 2011*). The value of $\Delta E$ was then calculated numerically from the height function, $h$, by triangulating $h$ and extracting the mean curvature and enclosed volume of the resulting mesh using pre-existing MATLAB (Mathworks, Natick, MA) software (*Mecklai, 2004*; *Kroon, 2014*; *Suresh, 2010*). To find the energy-minimizing filament-membrane conformation, $\Delta E$ was numerically computed while varying the deformed filament curvature, $1/R$, the size of $\Omega$, and the size of, and the filament height in, $U$ (the size of $U$ corresponds to the bound length of the filament; consistent with the discussion below, we find that $\ell_b = L_f$ in all cases of interest). As the simulation details above assume a perpendicular binding orientation relative to the long axis of a cylinder, for simplicity we model binding to deviatory angles by perpendicular binding to a different cell radius, $R$, where $\theta = \cos^{-1}(R/R_{\text{cell}})$. These implementation details were used to generate *Figure 2—figure supplement 1b–c* and *Figure 4—figure supplement 4*.

## 1.3. Preferred orientation of filament binding and binding phase diagram

In this section, following *Hussain et al. (2018)*, we provide details of analytic calculations complementing the numerical calculations discussed above. For a cylindrical membrane whose radius is larger than the radius of curvature of a filament, it is energetically favorable for the filament to bind at an angle of $\theta = 90°$ relative to the long axis of the cell, as this orientation requires minimal bending of both the protein and the membrane. For deviatory angles, $|\theta - 90°| > 0°$, an effective correction to the cell radius $R_{\text{cell}}$ is a multiplicative factor of $1/\cos\theta$, which makes binding less energetically favorable. The energetic penalty for MreB filaments binding at deviatory angles is approximately $40\ kT$ for a broad range of membrane pressures and depending on the membrane radius (below and *Figure 2B* of the main text). In general, higher membrane pressures make it more energetically favorable for the protein filament to bend, which minimizes the amount of volume displaced by the protein-membrane interaction (see below) (*Hussain et al., 2018*), while smaller membrane pressures make it more energetically favorable for the membrane to bend. Since a filament can always bend to conform to the membrane curvature, we see that large pressure differences across the membrane may enhance the energetic preference of an MreB filament for the perpendicular orientation.

We may summarize our results over a wide range of parameter values with an approximate phase diagram, assuming a cylindrical membrane (*Figure 2C* in the main text). We use the membrane pressure $p$ and the filament Young modulus $Y$, which varies with the bending energy $E_{\text{bend}}$ of the filament, as order parameters. By considering only the volume and surface height displaced directly underneath the binding region of the membrane, the deformation energy of the binding region, $U$, is (*Hussain et al., 2018*)

$$E_{\text{def},U} \approx \min_R \left[ \frac{B}{2}\left(\frac{1}{R} - \frac{1}{R_s}\right)^2 \ell_b + \frac{\pi b k_b \ell_b}{r_f} + \frac{p r_f \ell_b^3}{12}\left(\frac{1}{R} - \frac{1}{R_{\text{cell}}}\right) \right], \tag{S10}$$

where $b$ is the fraction of the filament cross-section needed to adhere to the membrane. The critical pressure at which the deformation is dominated by filament bending can be estimated as the value that sets $E_{\text{def},U}$ to be minimal at $R = R_{\text{cell}}$:

$$p^* \approx \frac{12B}{\ell_b^2 r_f}\left(\frac{1}{R_s} - \frac{1}{R_{\text{cell}}}\right), \tag{S11}$$

which, for the parameter values summarized in *Supplementary file 1*, estimates $p^* \approx 20$ kPa (*Hussain et al., 2018*). This value of $p^*$ is smaller than estimates of the turgor pressures of both *B. subtilis* and *E. coli* (*Supplementary file 1*), suggesting that in vivo, MreB

filaments always bend to adhere to the inner membrane. *Equation (S10)* is used under this assumption to generate the curves in *Figure 2B* of the main text. Furthermore, the linear dependence on $\ell_b$ of the first term of *Equation (S10)* implies that MreB filaments bind fully along their lengths. If $p < p^*$, as is the case for vesicles, then both the membrane and the MreB filament can deform each other in a manner that minimizes the total energy, with the membrane shape determined by *Equation (S6)*. For a large range of membrane pressures $0 \leq p \leq p^*$, we find that bound MreB filaments induce membrane curvature, and for vesicles where the pressure difference across the membrane is vanishingly small, the shape equation also predicts that MreB filaments can grossly deform the membrane (*Figure 2—figure supplement 1c*), a prediction consistent with experimental observations (*Salje et al., 2011*; *van den Ent et al., 2014*). For a filament with fixed dimensions, both $E_{\text{bend}}$ and the expression for $p^*$ in *Equation (S11)* are proportional to $Y$. Hence, $E_{\text{bend}} \propto p^*$ as $Y$ increases, and this relation is shown as the diagonal line in *Figure 2C* of the main text.

Similarly, considering only the membrane curvature induced by the binding region $U$, the filament does not bend if $E_{\text{bend}} + E > E_{\text{int}}$, where $E = \pi b k_b L_f / r_f$. If this inequality were satisfied, then the interaction energy $E_{\text{int}}$ may be too small to justify membrane binding, which requires a combination of polymer and membrane bending. Note that, while we assume $b = 1/6$ in this work, our results do not significantly change for different $b$, as shown in Figure 4 of (*Hussain et al., 2018*), and the regimes delineated in this section are summarized in *Figure 2C* of the main text.

Finally, we note that, for the parameter values summarized in *Supplementary file 1*, the prediction that it is energetically favorable for MreB filaments to align along the circumferential direction of a rod-like cell is robust in the case where the intrinsic curvature varies with position along the filament. In particular, a calculation based on *Equation (S2)* shows that this conclusion follows given that the filaments are, on average, more curved than the membrane. To see this, let $\kappa_s(\ell)$ denote the intrinsic filament curvature as a function of position along the filament length, $\ell$, and $\kappa$ denote the deformed curvature. In live cells, $\kappa$ does not vary as a function of $\ell$ because it is most energetically favorable for the filament to bend completely to match the ambient membrane curvature, as we have shown above. The total bending energy of the filament is then

$$E_{\text{bend}} = \frac{B}{2} \int_0^{L_f} (\kappa - \kappa_s(\ell))^2 d\ell. \tag{S12}$$

When binding to an angle that deviates from the circumferential direction in a cylinder, the deformed curvature will be smaller: let $\kappa = \kappa_0 - \kappa'$, where $\kappa_0$ is the curvature along the circumferential direction of a cylinder and $\kappa' \geq 0$ is a constant correction to $\kappa_0$ depending on the binding angle. Then, the difference in bending energies between binding in the direction of $\kappa$ as opposed the circumferential $\kappa_0$ direction is

$$\frac{B}{2} \int_0^{L_f} (\kappa')^2 d\ell + 2\kappa' \frac{B}{2} \int_0^{L_f} (\kappa_s(\ell) - \kappa_0) d\ell, \tag{S13}$$

which is larger than zero provided the filament is, on average, more curved than the cell (that is, the second term above is non-negative). Hence, because the binding orientation is robust, our model predictions will remain the same. Nevertheless, we note that cryo-EM experiments (*Hussain et al., 2018*; *Salje et al., 2011*; *van den Ent et al., 2014*) support the assumption of a uniformly bent MreB filament, and we have therefore focused on this case in the main text.

In the foregoing argument, we have assumed that $\kappa$ does not vary as a function of $\ell$. The case in which $\kappa$ varies significantly as a function of $\ell$, which is relevant to the geometry considered in *Figure 3F* of the main text, is more delicate. Note, however, that in this geometry $\kappa(\ell)$ can be explicitly calculated to show that MreB still orients in the largest principal direction. We consider, in particular,

$$E_1(v) = \frac{B}{2}\int_{v-d}^{v+d}(\kappa_1(\ell) - \kappa_s)^2 d\ell \text{ and } E_2(v) = \frac{B}{2}\int_{v-d}^{v+d}(\kappa_2(\ell) - \kappa_s)^2 d\ell, \tag{S14}$$

where, at an axial coordinate $v$, $E_1$ is the bending energy of a (constantly curved) filament aligning in the axial direction (with principal curvature $\kappa_1$) and $E_2$ is the bending energy of a filament aligning in the circumferential direction (with principal curvature $\kappa_2$). Here the integration is over the axial coordinates corresponding to filament length, $[v - d, v + d]$. *Figure 3—figure supplement 2c–d* shows that, in this case, binding along the greatest principal direction generally still incurs the least bending energy.

## 2. Curvature-based translocation of filaments

Consider a surface parameterized by $\mathbf{r} = r(u, v) \subset \mathbb{R}^3$, with $(x_1, x_2) = (u, v) \in \mathbb{R}^2$, which, for simplicity, we assume to be smooth almost everywhere so that the following quantities are well defined. The considerations below can be readily extended to the case of a piecewise smooth surface, as discussed in the following section (§2.1). We model a curved filament as a point $\mathbf{p}$ on this surface which translocates in a direction $\mathbf{d} \in \mathbb{R}^3$. At $\mathbf{p}$, the principal direction, $\mathbf{w}$, represented in the basis of the tangent space satisfies

$$S\mathbf{w} = c\mathbf{w}, \tag{S15}$$

where $S$ is the shape operator at $\mathbf{p}$ and $c$ is one of the principal curvatures of $\mathbf{r}$, hereafter taken to be the largest. Throughout this work, we assume the sign convention that the curvature is positive when any normal vector at $\mathbf{p}$ points towards the interior of $\mathbf{r}$, so that, for instance, the largest principal curvature is always positive for a cylinder (§2.2). When the filament translocates in the direction of the largest principal curvature, $\mathbf{d}$ satisfies $\mathbf{d} = \mathbf{w} \cdot (\mathbf{r}_u, \mathbf{r}_v)$, where $\mathbf{r}_u = \partial_u \mathbf{r}$ and $\mathbf{r}_v = \partial_v \mathbf{r}$.

For convenience, we recall the following statements from the main text. We set $\eta = \cos^{-1}\frac{\mathbf{d}\cdot\mathbf{r}_\theta}{\|\mathbf{d}\|\cdot\|\mathbf{r}_\theta\|}$, where $\eta$ is an angular deviation from the largest principal direction on the surface introduced by possible sources of stochasticity, the modified direction corresponds to an angle $\theta$ relative to the $u$-axis in parametric coordinates, $\mathbf{r}_\theta \in \mathbb{R}^3$ is the derivative of $\mathbf{r}$ in the direction of $\theta$, and distances are defined by the surface metric. Note that, when the largest principal direction is well defined, it is independent of the parameterization up to a sign (*Lee, 2009*). After the filament orients, it may translocate in the direction of $\mathbf{d}$ for a certain distance, dependent on the filament speed, before reorienting its direction of motion. As $\mathbf{d}$ is a vector in $\mathbb{R}^3$, arbitrarily moving in the direction of $\mathbf{d}$ may, however, move the filament out of $\mathbf{r}$. To define the translocation consistently, we define it $(u, v)$-space by taking the translocation angle with respect to the $u$-axis to also be $\theta$. Translocating along an angle $\theta$ with respect to the $u$-axis in $(u, v)$-space ensures that the filament remains on the surface, and the direction of translocation corresponds to that on the smooth patch defined locally on $\mathbf{r}$.

As a discrete-time flow in parametric space, and with suitable units of time so that the filament may reorient at every timestep, the 2D equation of filament motion is

$$X_{n+1} = X_n + \chi_n \ell_n (\cos\theta_n, \sin\theta_n), \tag{S16}$$

where, in $(u, v)$-space, $X_n$, $\ell_n$, and $\theta_n$ are the position, step size, and translocation angle, respectively, of the filament at a timestep $n$. Here $\theta_n$ is the value of $\theta$ computed at the surface point corresponding to $X_n$ and assuming $\eta \sim \mathcal{N}(0, \sigma^2)$—that is, the angular noise is normally-distributed, with mean zero and variance $\sigma^2$. The strength of the noise may depend on factors such as the energetic difference of binding along the two principal directions, and we examine cases where $\sigma$ depends on the difference, $\Delta c$, between principal curvatures below. $\chi_n$ is a random sign, which accounts for the possibility of both left-handed and right-handed translocation, and may not substantially vary in $n$ if filament motion is processive (namely, the filament does not backtrack). We assume that $\ell_n$ satisfies the integral equation

$$L = \int_0^{\ell_n} \sqrt{g_{11}(X(\tau))\cos^2\theta_n + 2g_{12}(X(\tau))\cos\theta_n\sin\theta_n + g_{22}(X(\tau))\sin^2\theta_n}\,d\tau, \tag{S17}$$

where $X(\tau) = X_n + \tau(\cos\theta_n, \sin\theta_n)$ and $g$ denotes the metric tensor (with $u$ and $v$ corresponding to the indices 1 and 2, respectively), which relates it to a constant filament step size, $L$, on the surface. When $g$ is slowly varying, that is $g(X(\tau)) \approx g(X_n)$, as is expected in the limit of small $\ell_n$, or in particular

$$\frac{\ell_n \hat{Y} \cdot \nabla I(X_n)}{4I(X_n)} \ll 1, \tag{S18}$$

where $\hat{Y} = (\cos\theta_n, \sin\theta_n)$, $I$ is the integrand of **Equation (S17)**, and we discard higher-order terms in $\nabla I$ and $\ell_n$, **Equation (S17)** simplifies to a linear equation for $\ell_n$:

$$L = \ell_n \sqrt{g_{11}(X_n)\cos^2\theta_n + 2g_{12}(X_n)\cos\theta_n\sin\theta_n + g_{22}(X_n)\sin^2\theta_n}. \tag{S19}$$

In general, we wish to determine the distribution of $X$ in $(u, v)$-space, which would determine the distribution of the filament on the surface. Below, we introduce activation and deactivation dynamics, as discussed in the main text, and examine the statistics of $X$ on several surfaces.

## 2.1. Numerical solutions in the finite and continuum cases

Before discussing the implications of **Equation (S16)**, we provide implementation details for the simulations and numerical calculations discussed below and in the main text. We numerically implemented the dynamics of **Equation (S16)** with Langevin simulations and verified our results with coarse-grained, continuum calculations corresponding to solutions of the corresponding Fokker-Planck equation. For the former, individual filaments were activated and deactivated as discussed in the main text, and their trajectories were simulated directly according to **Equation (S16)**. The final positions of all filaments were pooled together to determine the corresponding filament concentrations. For the latter, we discretized $(u, v)$-space uniformly into $M_u \times M_v$ rectangular elements, computed the transition matrix corresponding to **Equation (S16)** and a given set of parameter values, and multiplied a vector corresponding to the filament number, $N_F$, to model the dynamics. Deactivation effects were modeled by multiplying the vector of filament numbers after each timestep by $e^{-\lambda}$, and filament concentrations were determined by dividing the vector of filament numbers by the corresponding surface area elements. Specific implementation details, such as geometric parameters, are summarized for each figure in this work in §3.

For a piecewise parametric surface, we allowed filaments to translocate between two subsurfaces while conserving the total step size, $L$, translocated at every timestep. For the piecewise geometries considered in this work, the filament coordinates were connected between subsurfaces in an obvious manner.

## 2.2. Cylinder

Consider a cylinder parameterized by $\mathbf{r} = (a\cos u, a\sin u, av)$, where $a > 0$, $u \in [0, 2\pi]$, $v \in [0, z]$, and $z$ is a large enough constant so that we do not consider filament translocation out of the cylinder. The nonvanishing components of the metric tensor are $g_{11} = g_{22} = a^2$. Hence, the step sizes in parameter space are identical at all timesteps $n$ and equal to $\ell_n = \ell = L/a$, and translocation occurs mainly along the $u$-axis. Assuming processive filament motion, we recover a Pearson-like random walk in a 2D plane with periodic boundary conditions in $u$, where the translocation angle satisfies $\theta_n \sim N(0, \sigma^2)$, and $\chi_n = \pm 1$ for all $n$. Correlated Pearson random walks of a similar form have been studied by Kareiva and Shigesada in the context of insect movement (**Kareiva and Shigesada, 1983**). Denoting by $U_N$ and $V_N$ the displacement along the $u$ and $v$ coordinate, respectively, after $N$ steps, with $U_N = \sum_{i=1}^N \ell\cos\theta_i$ and $V_N = \sum_{i=1}^N \ell\sin\theta_i$, we find that

$$\langle U_N \rangle = \ell N e^{-\sigma^2/2}, \quad \langle V_N \rangle = 0, \quad \mathrm{Var}(U_N) = \ell^2 N e^{-\sigma^2}(\cosh(\sigma^2)-1),$$
$$\mathrm{Var}(V_N) = \ell^2 N e^{-\sigma^2}\sinh(\sigma^2), \tag{S20}$$

in agreement with simulations of *Equation (S16)* for parameter values relevant to *B. subtilis* (§3.1 and *Figure 3B* of the main text).

## 2.3. Torus and Helix

Let $\mathbf{r} = ((R + a\sin u)\cos(av/R), (R + a\sin u)\sin(av/R), a\cos u)$, where $u \in [0, 2\pi]$, $v \in [0, R\Phi/a]$, $\Phi$ is the axial subtended angle, and $0 < a < R$. Here the nonvanishing elements of the metric tensor are $g_{11} = a^2$ and $g_{22} = a^2(1 + (a/R)\sin u)^2$, and the step sizes depend on $u$. As the principal curvatures in the circumferential $u$-direction, $c_u$, and in the axial $v$-direction, $c_v$, satisfy $c_u = 1/a > c_v = (R + a\sin u)^{-1}\sin u$, translocation occurs mainly in the direction of $u$. Due to the dependence of the step sizes on $u$, $\langle U_N \rangle$, $\langle V_N \rangle$, $\mathrm{Var}(U_N)$, and $\mathrm{Var}(V_N)$ may differ from that of a cylinder and are analytically difficult to calculate. We focus instead on determining the filament concentration, $C_F$, which, unlike the case of a cylinder, will be non-uniform in $X$.

For intuition, we consider the case of no translocation noise. Consider the probability $P_N(X;Y)$ of observing the filament at any point $X$ in $(u,v)$-space at a timestep $N$ given that it is initially at $Y$, and assume $p_{\mathrm{init}}(Y)$ to be proportional to $dA = a^2(1 + (a/R)\sin u)dudv$. Since $dA$ depends on $u$, the probability $P_N(X)$ of observing the filament at a timestep $N$, averaged over initial positions, is not uniform. Nevertheless, considering the dynamics of activating and deactivating filaments as above, we find that, in the limit of continuous time, the expected number of filaments at a coordinate $X = (u,v)$ and time $t$ is

$$\begin{aligned} N_F(X,t) &= \int_0^t [dA(u - \tau\nu, v) + dA(u + \tau\nu, v)]k'e^{-\lambda\tau}d\tau \\ &= 2a^2k'\left(\frac{1-e^{\lambda t}}{\lambda} + \frac{a\lambda\sin u - a\lambda e^{-\lambda t}\sin u\cos(t\nu) + a\nu e^{-\lambda t}\sin u\sin(t\nu)}{R(\lambda^2 + \nu^2)}\right), \end{aligned} \tag{S21}$$

where $k' = kp_{\mathrm{init}}/dA$ and $\nu$ is a speed corresponding to $U_N$ in the continuum limit. As $\lambda \to \infty$, corresponding to the limit in which filaments are instantaneously deactivated, *Equation (S21)* predicts a number enhancement on the outer edge of the form $N_F(X,t) = (2a^2k'/\lambda)(1 + (a/R)\sin u)$, while as $\lambda \to 0$, corresponding to the limit in which filaments persist indefinitely, *Equation (S21)* predicts an approximate uniform distribution $2a^2k'(t + \frac{a}{R\nu}\sin(t\nu)\sin u) \approx 2a^2k't$ in the limit of large $t$. While the concentration of filaments, $C_F$, is uniform over the surface in the former case, the latter case corresponds to the 'washing-out' of initial activation conditions and implies a larger value of $C_F$ on the inner edge. Namely, in the formal limit $\lambda \to 0$ followed by $t \to \infty$, with $\lambda t \to 0$,

$$\frac{C_F(X,t)}{t} \to \frac{2k'}{(1 + (a/R)\sin u)}, \tag{S22}$$

which is maximized at $u = -\pi/2$, corresponding to the inner edge of the torus. The filament concentration at the inner edge becomes larger than that of the outer edge by a factor of $(1 + a/R)/(1 - a/R)$, which depends only on the geometry of the torus. This effect may be interpreted as a 'geometric focusing' caused by both the filament number, $N_F$, becoming uniform in $(u,v)$-coordinates and variations in the area element $dA$ in $u$ or $v$. Even for a range of noises, including the limit of large noise ($\sigma \to \infty$), simulations show that this description is valid and agrees with *Equation (S22)* (*Figure 4—figure supplement 1a*).

While the geometric focusing effect is clearly applicable to other geometries, particularly those involving bent or sinusoidal surfaces (*Takeuchi et al., 2005*; *Renner et al., 2013*), we examine whether it applies to ellipsoids and other geometries considered in this work below. We also note that this effect is different from filaments translocating along a direction of curvature until it is 'attracted,' as is the case for a spherocylinder (§2.5) or a surface with small wavelength undulations (*Figure 3F* of the main text and §2.8). The main difference is that the dynamics corresponding to geometric focusing is recurrent, and hence, due to spatial variations in the area element $dA$, leads to $C_F$ depending on geometric features such as in

*Equation (S22)*. In contrast, the dynamics corresponding to an attractor is not. For instance, in the absence of noise and the limit of large processivity, $C_F$ is nonzero over a toroidal surface and the relative localization is determined by *Equation (S22)*, but $C_F$ vanishes at the hemispherical caps on a spherocylinder, as discussed later (§2.5).

Note that characteristic values of MreB persistence, as summarized in *Supplementary file 2*, suggest that MreB filaments lie in the $\lambda \approx 0$ regime. The observation of enhanced MreB concentration at the inner edge of toroidal cells is quantitatively consistent with the geometric focusing effect discussed in this section, as explained in the main text and prior work (*Wong et al., 2017*).

Finally, we note that similar arguments as that above describe the case of a helix, which can be parameterized by

$$\mathbf{r} = \left( (R + a\sin u)\cos(av/R) + \frac{\phi a\sin(av/R)\cos u}{\sqrt{R^2 + \phi^2}}, (R + a\sin u)\sin(av/R) \right.$$
$$\left. -\frac{\phi a\cos(av/R)\cos u}{\sqrt{R^2 + \phi^2}}, \frac{\phi av}{R} + \frac{aR\cos u}{\sqrt{R^2 + \phi^2}} \right),$$

(S23)

where $u \in [0, 2\pi]$, $v \in [0, R\Phi/a]$, $\Phi$ determines the extended length of the helix, $0 < a < R$, and $\phi$ is the helical pitch. Here the nonvanishing elements of the metric tensor are $g_{11} = a^2$, $g_{12} = g_{21} = \phi a^3 (R\sqrt{R^2 + \phi^2})^{-1}$, and $g_{22} = a^2 (2(R^2 + \phi^2)^2 + a^2(R^2 + 2\phi^2) + aR(4(R^2 + \phi^2)\sin u - aR\cos 2u))(2R^2(R^2 + \phi^2))^{-1}$, and the step sizes again depend on $u$. As the principal curvatures in the circumferential direction, $c_u$, and in the axial direction, $c_v$, satisfy $c_u = 1/a > c_v = (R^2 + \phi^2 + aR\sin u)^{-1}R\sin u$, translocation occurs mainly in the direction of $u$. Like the case of a torus, we expect filaments to become uniformly distributed across circumferential hoops in the limit of infinite processivity. The predicted value of $C_F$ in this case, $C_F \propto 1/dA = (a^2(R^2 + \phi^2 + aR\sin u)dudv)^{-1}R\sqrt{R^2 + \phi^2}$, is quantitatively consistent with numerical simulations and larger at the inner edge (*Figure 4—figure supplement 1b–c*).

## 2.4. Ellipsoid

Consider the surface $\mathbf{r} = (a\sin u\cos v, a\sin u\sin v, b\cos u)$, where $a, b > 0$, $u \in [0, \pi]$, and $v \in [0, 2\pi]$. Here the nonvanishing components of the metric tensor are $g_{11} = (a^2 + b^2 + (a^2 - b^2)\cos(2u))/2$ and $g_{22} = a^2\sin^2 u$, and $dA = \frac{1}{\sqrt{2}}a\sin u\sqrt{a^2 + b^2 + (a^2 - b^2)\cos(2u)}dudv$. When $b > a$, the principal curvatures in the circumferential $v$-direction, $c_v$, and in the axial $u$-direction, $c_u$, satisfy

$$c_v = \frac{\sqrt{2}b}{a\sqrt{a^2 + b^2 + (a^2 - b^2)\cos(2u)}} \geq c_u = \frac{2\sqrt{2}ab}{(a^2 + b^2 + (a^2 - b^2)\cos(2u))^{3/2}},$$

(S24)

with equality only at the poles ($u = 0, \pi$). Hence, in the case of no noise, filaments always translocate in the direction of $v$, and similarly, the opposite is true when $b < a$.

As calculations involving a finite noise are analytically complex, we first consider the case of no noise. Here $U_N = 0$ and $V_N^u = LN/(a\sin u)$, where $U_N$ is the displacement along the $u$-coordinate after $N$ steps and $V_N^u$ is the displacement along the $v$-coordinate after $N$ steps assuming that the filament remains on the curve of constant $u$. For an ensemble of filaments, considering similar activation and deactivation dynamics as above gives

$$N_F(X,t) = \int_0^t \left[ dA\left(u, v - \tau\frac{\nu}{\sin u}\right) + dA\left(u, v + \tau\frac{\nu}{\sin u}\right) \right] k' e^{-\lambda\tau} d\tau \tag{S25}$$

$$= \frac{\sqrt{2}k'(1-e^{-\lambda\tau})}{\lambda} a\sin u \sqrt{a^2 + b^2 + (a^2 - b^2)\cos(2u)} \tag{S26}$$

$$= \frac{2k'(1-e^{-\lambda\tau})}{\lambda} \times \frac{dA}{dudv}, \tag{S27}$$

where $k'$ is the same as above and $\nu$ is a constant speed corresponding to $V_N = V_N^u \sin u$ in the continuum limit. Hence, for any value of $\lambda$, $C_F$ is a constant, in agreement with numerics (**Figure 4—figure supplement 2**).

We next consider the case of a finite noise. We first note that the area element $dA$ vanishes at the tips, and here the resulting value of $C_F$ would diverge if $N_F$ were nonzero. To compare filament concentrations over regions with differently sized area elements, it is convenient to define the *average concentration* on the (sub)surface $S$ as

$$\langle C_F \rangle_S = \frac{\sum_{X \in S} N_F(X)}{\sum_{X \in S} dA(X)}, \tag{S28}$$

which is different from a direct averaging of $C_F$ over area elements as considered for a torus (the latter has the form of a harmonic sum). Numerical calculations accounting for an ensemble of filaments with similar activation and deactivation dynamics as above show that, intriguingly, $\langle C_F \rangle_S$ is enhanced at the ellipsoidal poles ($S = \{X : u < \pi/4 \text{ or } u > 3\pi/4\}$) for a range of noises (**Figure 4—figure supplement 2**). Different from the case of zero noise, here the increased values of $C_F$ at the poles arise because filaments may randomly translocate to the poles and the area element is significantly smaller there than that away from the poles. Similar observations also hold in the limit of large noise, $\sigma \to \infty$, and the limiting value of the ratio of $\langle C_F \rangle_S$ at the tips versus the bulk is the ratio of the corresponding areas (**Figure 4—figure supplement 2**).

In the case of a finite noise, localization in a torus, a helix, and an ellipsoid arises because of $N_F$ becoming spatially homogeneous and spatial variations of the area elements. However, in the case of zero noise, we remark on a key difference between $C_F$ on a torus and an ellipsoid as follows. For a torus, spatially heterogeneous filament concentration arises due to the trajectories of individual filaments being closed orbits in the circumferential direction and the long persistence of filaments, which 'washes out' the initial distribution of filament position. For an ellipsoid, filaments do not move in the axial direction and the resultant filament concentration remains uniform on the surface, regardless of persistence. That a uniform distribution is maintained in the absence of noise on an ellipsoid is also different from the cases of a spherocylinder (§2.5) and a surface with small wavelength undulations (**Figure 3F** of the main text and §2.8), as discussed later. In general, $C_F$ would be uniform in the case of zero noise for any surface where the direction in which $dA$ varies does not coincide with the direction of filament translocation.

## 2.5. Spherocylinder

Consider the surface parameterized piecewise by $\mathbf{r}_1 = (a\cos u, a\sin u, av)$, where $a > 0$, $u \in [0, 2\pi]$, $v \in [0, z]$ for some $z > 0$, $\mathbf{r}_2 = (a\sin v\cos u, a\sin v\sin u, a\cos v + az)$, where $u \in [0, 2\pi]$ and $v \in [0, \pi/2]$, and $\mathbf{r}_3 = \mathbf{r}_2 - (0, 0, az)$, where $u \in [0, 2\pi]$ and $v \in [\pi/2, \pi]$. In the absence of noise, filaments translocate along the $u$ axis in the cylinder and a random angle in the hemispheres. This could allow for translocation out of the caps and into the cylindrical body. Indeed, in the absence of noise, it is clear that any filament in the hemispherical caps will eventually translocate into and remain at the cylindrical rims.

For the spherocylindrical surface described above, general calculations in the case of a finite noise are analytically complex due to the irregular geometry. In the presence of noise, however, the depletion of filaments at the hemispherical caps can be supported numerically. For parameter values relevant to MreB in *E. coli* and *B. subtilis* (**Supplementary file 2**), numerical calculations show that $\langle C_F \rangle_S$ is larger by a multiplicative

factor of ~2.0 when $S$ is the cylindrical bulk than that when $S$ is a hemispherical endcap (**Figure 3C** of the main text).

## 2.6. Filament concentration is independent of Gaussian curvature

While filament movement depends on the principal curvatures, we wondered whether this implies that filament concentration always depend on the Gaussian curvature. Here we show that, for general surfaces, filament concentration is independent of Gaussian curvature under the dynamics considered in this work. Consider the parameterization $\mathbf{r} = (\sin u, (1 - \cos u) \cos u, v)$, where $u \in [-\pi, \pi]$, $v \in [0, z]$, and $z$ is a large enough constant so that we do not consider filament translocation out of the surface. Although the Gaussian curvature vanishes everywhere, any cross-section of the surface has regions of both negative and positive principal curvature, and the regions of positive principal curvature represent attractors to the filament dynamics: filaments translocating into such regions rarely translocate out.

While numerical simulations show that filaments are localized at regions where the largest principle direction coincides with the axial direction (**Figure 4C** in the main text and **Figure 4—figure supplement 3a–b**), analytical calculations are difficult to undertake due to the complicated geometry. Nevertheless, we may consider a similar 1D problem of a particle moving with velocity $\nu$ along a circular ring, which is parameterized without loss of generality by $u \in [0, 2\pi]$, and contains a single absorbing coordinate at $u = 0$. Accounting for activation and deactivation dynamics as above, the filament number at $u = 0$ at a time $t > t^*$ in the case of vanishing translocation noise ($\sigma = 0$) and assuming $\nu > 2\pi\lambda$, so that the flux described below is nonzero, can be written as

$$N_F(u = 0) = \int_0^t \left( \int_{\max(t_1 - t^*, 0)}^{t_1} k' dA(u - \nu(t_1 - t_0), v) e^{-\lambda(t_1 - t_0)} dt_0 \right) e^{-\lambda(t - t_1)} dt_1, \qquad \text{(S29)}$$

where $k'$ is defined above, the term in parenthesis is the filament flux into $u = 0$ at time $t_1$, and $t^* = 2\pi/\nu$ represents the maximal time needed for any filament to become absorbed. Further assuming a constant value of $dA(u, v) = dA$ for simplicity, direct evaluation of **Equation (S29)** yields

$$N_F(u = 0) = \frac{k' dA}{\lambda^2} \left( 1 - e^{-\frac{2\pi\lambda}{\nu}} - \frac{2\pi\lambda}{\nu} e^{-\lambda t} \right), \qquad \text{(S30)}$$

which quantitatively describes the dependence of the filament number at the absorbing coordinate as a function of the processivity, which is determined by $\lambda$, and other parameters. Note that, as $\lambda \to 0$, corresponding to the case of infinite processivity, $N_F(u = 0)$ is predicted to diverge, while as $\lambda \to \infty$, corresponding to the case of zero processivity, $N_F(u = 0) \to 0$. It is straightforward to generalize **Equation (S30)** to the case of several absorbing points and different geometries, provided that similar simplifying assumptions can be employed as above. Importantly, these calculations can be compared to numerical calculations for the geometry considered in this section. While the value of $N_F$ at attracting regions may generally depend on the geometry of such regions, the dependence of $N_F$ on the processivity, $\lambda$, can be conveniently explored by defining the *localization ratio*, $\rho$, as the ratio between filament numbers at a certain value of $\lambda$ compared to that at $\lambda/2$:

$$\rho = \frac{N_F(u = 0, \lambda)}{N_F(u = 0, \lambda/2)}. \qquad \text{(S31)}$$

Defining $\rho$ analogously for the geometry considered in this section, we find that numerical calculations of $\rho$ for the geometry considered here across a range of $\sigma$ are consistent with the theoretical prediction for $\rho$ based on the simplified model considered in this paragraph (**Figure 4—figure supplement 3c**). Hence, we conclude that **Equation (S30)** captures the dependence of $N_F$ on $\lambda$ for more general geometries.

## 2.7. Cylinder with a bulge

Building on work that has examined MreB motion in cells with membrane bulges (**Hussain et al., 2018**), here and below we consider bulges of positive Gaussian curvature on a cylindrical surface. We consider a class of cylinders with bulges parameterized piecewise by $\mathbf{r}_1 = (v, \cos u, \sin u)$, where $u \in [0, 2\pi]$ and $v \in [0, z]$ for some $z > 0$ and $\sqrt{(u - \pi/2)^2 + (v - z/2)^2} < b$ and $\mathbf{r}_2 = (b \sin v \cos u + z/2, b \sin v \sin u, sc \cos v + 1)$ where $u \in [0, 2\pi]$ and $v \in [0, \pi/2]$ otherwise. We suppose $b, c > 0$ and $b < z$, so that the intersection area constitutes a small fraction of the cylindrical body. Here $s = 1$ if the bulge protrudes outward and $s = -1$ if the bulge protrudes inward. For this class of parametric surfaces, three general cases of outward bulges ($s = 1$) can be classified depending on whether $c = b$, $c < b$, or $c > b$. Upon computing the principal curvatures in each case, we find that the three cases correspond respectively to *random translocation* ($c = b$), in which case the translocation direction is random; *polar translocation* ($c < b$), in which case translocation occurs predominantly along the bulge $v$-coordinate; and *circumferential translocation* ($c > b$), in which case translocation occurs predominantly along the bulge $u$-coordinate (**Figure 3—figure supplement 1a**). We note that inward bulges of positive Gaussian curvature ($s = -1$) exhibit the same behavior, with polar translocation when $c > b$ and circumferential translocation when $c < b$.

**Figure 4B** of the main text illustrates filament dynamics for both random and circumferential translocation where $b = c = 0.5$ (spherical bulge) and $b = 0.5, c = 1$ (ellipsoidal bulge), in both the cases of zero and infinite processivity. We focus here on the latter geometry as the motion there is consistent with experiments (**Hussain et al., 2018**). When there is no noise, considering similar activation and deactivation dynamics as above shows that, similar to a spherocylinder (§2.5), the bulge attracts filaments in the case of small $\lambda$ or large processivity. As simulations demonstrate, this is also true in the case of a finite noise, where we find the bulge to contain larger numbers of filaments relative to the case of a uniform distribution on the surface (**Figure 4B** in the main text). For characteristic parameter values relevant to *B. subtilis* MreB, we find that $\langle C_F \rangle_S$ is increased at the bulge (**Figure 3E** of the main text) and substantial localization may occur at the bulge neck, consistent with previous experiments (**Figure 3—figure supplement 1b**).

## 2.8. Cylinder with small wavelength undulations

Previous work has examined the correlation of MreB concentration with subcellular-scale shape fluctuations in *E. coli* cells (**Ursell et al., 2014**). Based on the weak correlation observed between outer and inner contour curvatures in **Ursell et al. (2014)**, the authors determined that the experimentally observed MreB enrichment was not caused by bending modes: in a circular torus, for instance, the correlation between outer and inner contour curvatures should be strictly negatively correlated. The authors concluded that short length-scale, high magnitude fluctuations dominate experimental observations of cell shape.

To probe such a geometry, we consider the parameterization $\mathbf{r} = ((a + c \sin(Pv)) \sin u, (a + c \sin(Pv)) \cos u, v)$, for $u \in [0, 2\pi]$ and $v \in [0, 2\pi]$. Here $a$ denotes an average cylinder radius, we assume that $0 < c \ll a$, and $P > 0$ is a variable controlling the number of periods. For large wavelength undulations, $P$ is less than, or on the order of, unity. However, the Gaussian and mean curvatures are uncorrelated in this case, inconsistent with the positive correlation observed in **Ursell et al., 2014** for cells growing in sinusoidally-shaped chambers, and the ranges of Gaussian and mean curvatures are significantly smaller than those measured for wild-type, unconfined, sinusoidally-confined, thin, and wide cells in different experiments (**Ursell et al., 2014**; **Shi et al., 2017**; **Bratton et al., 2018**) (**Figure 3—figure supplement 2a** and **Figure 3G** of the main text). We therefore consider a geometry with short wavelength undulations, for which $P \geq 1$ (**Figure 3—figure supplement 2b**). Numerical results for principal curvature-dependent translocation on this geometry, which is consistent with the predicted filament binding orientation (**Figure 3—figure supplement 2c–d**), are presented in **Figure 3G** in the main text.

## 2.9. Effects of principal curvature-dependent translocation noise and varying filament step size on model predictions

Prior experiments have shown that the variation in MreB trajectory direction is width-dependent, suggesting that the translocation noise may depend on the difference of principal curvatures, $\Delta c$ (**Hussain et al., 2018**). As discussed in the main text, we may model this observation by letting $\sigma$ vary with $\Delta c$: for simplicity, we set

$$\sigma = \alpha(\Delta c)^{-1} \tag{S32}$$

(**Figure 3—figure supplement 3a**), but note that more complicated functional dependencies, such as a quadratic dependence of the form $\sigma = \beta(\Delta c)^{-2}$, do not significantly change our results (**Figure 3—figure supplement 3b**). Unless otherwise indicated (see §3), all simulations and calculations in this work pertaining to MreB assume **Equation (S32)**, and we further note that, for the parameter values relevant to MreB translocation in *E. coli* and *B. subtilis* summarized in **Supplementary file 2**, taking a constant value of $\sigma = 0.3$ also does not significantly change our results. Similarly, we verify that localization arises even for vary large step sizes ($L = 2~\mu\text{m}$), in which case MreB filaments realign infrequently (**Figure 3—figure supplement 3b**).

## 2.10. Effects of filament twist, flexural rigidity, and Gaussian curvature-dependent activation on model predictions

Recent work has shown that regions of negative Gaussian curvature can allow twisted filaments to bind with low elastic energy (**Quint et al., 2016**). In another study, Wang and Wingreen assumed that MreB assembles into bundles with significantly larger flexural rigidity than that of filaments considered in this work (**Wang and Wingreen, 2013**). In this section, we examine how our model predictions for MreB localization in *E. coli* change over ranges of three parameters: (1) the intrinsic filament twist, $\omega_0$, (2) the coupling, $\gamma$, of filament activation to Gaussian curvature, and (3) the filament flexural rigidity, $B$.

We first note that $\gamma$ may vary independently of $\omega_0$. Quantitatively, the rate of filament activation may not only depend on the parameters of the twist, but also on other cellular parameters (**Wong et al., 2017**). We assume that the filament activation rate per unit area, $k$, varies with $\gamma$ as

$$k(u,v) = k_0 - \gamma G(u,v), \tag{S33}$$

where $k_0$ and $\gamma$ are constants and $G(u,v)$ is the Gaussian curvature at the parametric coordinate $(u,v)$. When $\gamma = 0$, we recover our original assumption that the activation rate per unit area is constant.

We now explore the effects of twist and Gaussian curvature-dependent activation on filament concentration. For simplicity, we consider a torus (**Figure 4—figure supplement 1a**), for which $G = \sin u[a(R + a\sin u)]^{-1}$ (c.f. §2.3), but show later in **Figure 3—figure supplement 3b** that the results for the undulating geometry of **Figure 3F** of the main text also remain qualitatively similar. Due to twist, MreB filaments may move in a direction which deviates from the direction of largest curvature. In particular, while further work should verify the robustness of this equation for nonzero turgor pressures and intrinsic filament curvatures, Equation (12) of **Quint et al. (2016)**,

$$2B\frac{\sin^3\theta_0\cos\theta_0}{a^2} = K\frac{\cos(2\theta_0)}{2a}\left(\frac{\sin(2\theta_0)}{2a} + \omega_0\right), \tag{S34}$$

predicts the angular deviation from the largest principal direction in a cylinder, $\theta_0$, due to twist in the limit that MreB filaments are fully bound to the membrane. Here $B$ is the flexural rigidity of a filament, $K$ is the elastic twist stiffness, $a$ is the cell radius, and $\omega_0$ is the intrinsic helical twist. Thus, upon knowing the values of $k_0$, $\omega_0$, $\gamma$, and $B$, we may compute $k$ and $\theta_0$ using **Equation (S33) and (S34)**, from which we can determine $C_F$ via simulations similar to those

above. Doing so for a large parameter range which includes characteristic values of $B$ (**Supplementary file 1**) and theoretically hypothesized values for MreB of $K = 2000 \, kT \cdot \mathrm{nm}$ and $\omega_0 a = 1$ to $5$ (**Quint et al., 2016**) reveals the final ratio of filament concentrations to be quantitatively similar in all cases (**Figure 4—figure supplement 4a–f**). We note, in particular, that the effect of filament twist alone is small in all cases: this is because the biasing of translocation angles due to twist, as predicted by **Equation (S34)**, is irrelevant for the toroidal geometry, where translocating along deviatory angles still traces out hoops (c.f. §2.3). In these simulations, to illustrate the range of translocation behavior we have assumed that binding is always energetically favorable; however, there will be an energetic cost of unwinding of the form

$$E_{\mathrm{twist}} = \frac{K}{2} \int_0^{\ell_b} (\omega - \omega_0)^2 dl, \tag{S35}$$

where $\omega$ is the bound filament twist (**Quint et al., 2016**). For the parameter values considered in this work (**Supplementary file 1**), binding becomes energetically unfavorable at large twists ($\omega_0 a \gtrsim 100$), for which $\theta_0 \approx \pi/4$.

Additionally, our previous measurements of MreB fluorescence in bent *E. coli* cells confined to toroidal microchambers (which may be modeled as torii with geometric parameters $a = 1$ and $R = 10$) shows MreB concentration to be enhanced at the inner edges by a factor of approximately $1.1$ (**Wong et al., 2017**). This suggests the strength of Gaussian curvature coupling for MreB, if indeed such coupling does exist, to be small, as shown in **Figure 4—figure supplement 4g–h**. As we have demonstrated previously (**Wong et al., 2017**), the small observed enhancement is consistent with processivity alone. Furthermore, as mentioned above, modeling characteristic parameter values of both filament twist and Gaussian curvature-dependent activation still results in localization for geometries other than a torus, such as that considered in **Figure 4F** of the main text (**Figure 3—figure supplement 3b**).

Finally, we were interested to determine if our model predictions were robust to variation in the filament flexural rigidity, which for the parameter values in **Supplementary file 1** has a value of $B \approx 1.65 \times 10^{-25} \, \mathrm{J} \cdot \mathrm{m}$. To explore this for a torus, we repeated the foregoing dynamical simulations across a range of flexural rigidities. We first considered a ten-fold smaller value of $B = 1.0 \times 10^{-26} \, \mathrm{J} \cdot \mathrm{m}$, which, by **Equation (S10)**, still predicts (1) the filament to bend to conform to the membrane and (2) the depth of the potential well corresponding to **Figure 2B** of the main text to be approximately $3 \, kT$, a value that may be large enough to be robust to thermal noise and other sources of stochasticity (**Figure 4—figure supplement 4k**). We next considered a 100-fold larger value of $B = 1.5 \times 10^{-23} \, \mathrm{J} \cdot \mathrm{m}$, approximately the flexural rigidity of a thick bundle with $r_f = 10 \, \mathrm{nm}$ (**Wang and Wingreen, 2013**). For this value of $B$ and $r_f$, no twist, and the remaining parameters summarized in **Supplementary file 1**, **Equation (S11)** estimates the critical pressure to be $p^* \approx 5 \, \mathrm{atm}$, a value which is larger than characteristic estimates of turgor in *E. coli*. As the corresponding value of $E_{\mathrm{bend}}$ is less than $E_{\mathrm{int}}$, we anticipate that both the membrane and the bundle may bend (see **Figure 2C** of the main text). To show that it remains energetically favorable for thick bundles to bind to membranes and that the binding orientation along the largest principal direction is robust, we numerically solved the shape equation (**Equation (S6)**) and found that the free energy change due to binding, $\Delta E$, decreases with the size of the domain $\Omega$ and, for a given $\Omega$, is minimal when the membrane bends to conform to the bundle (**Figure 4—figure supplement 4k**). In the limit $\Omega \to U$, the Monge gauge assumption underlying **Equation (S6)** becomes invalid; nevertheless, $\Delta E$ tends to the analytical expression of **Equation (S10)** under the condition that the minimizer, $R$, is close to the intrinsic radius of curvature of the bundle, $R_s$. In both this limit and the simulations of **Figure 4—figure supplement 4k**, binding remains energetically favorable and the binding orientation remains robust even for thick filaments, suggesting that translocating along directions of largest principal curvature remains relevant. **Figure 4—figure supplement 4f,i–j** shows simulation results for both values of $B$ compared to the value ($1.65 \times 10^{-25} \, \mathrm{J} \cdot \mathrm{m}$) assumed in this work. We find that, in all cases, the model predictions remain quantitatively similar.

## 3. Summary of Figures

For convenience, here we summarize implementation details used to generate figures in this work.

### 3.1. *Figure 3B* of the main text

Here $a = 1$ and $z = 100$. Langevin simulations to generate trajectories were undertaken with $10^5$ activated filaments, $L = 0.4$, $\sigma = 0.3$, and $N$ determined by the number of steps needed to translocate one hoop. Filaments were activated at the center ($v = 0$) so that none of them translocated beyond the range specified by $z$.

### 3.2. *Figure 3C* of the main text

Here $a = 1$ and $z = 4$. Langevin simulations to generate a representative trajectory were undertaken with a single activated filament and the parameters relevant to *E. coli* summarized in **Supplementary file 2**, except $N = 500$. Note that we use the linear relation $\sigma = \alpha(\Delta c)^{-1}$, where $\Delta c$ is the difference of principal curvatures and the value of $\alpha$ is provided in **Supplementary file 2**. Numerical calculations for ensemble dynamics were undertaken with the parameters relevant to *E. coli* summarized in **Supplementary file 2**. Each subsurface is discretized into $60 \times 60$ bins.

### 3.3. *Figure 3D* of the main text

Here $a = 1$, $R = 10$, and $\Phi = \pi$. Periodic boundary conditions in $v$ are assumed. Note that we use the linear relation $\sigma = \alpha(\Delta c)^{-1}$, where $\Delta c$ is the difference of principal curvatures and the value of $\alpha$ is provided in **Supplementary file 2**. Numerical calculations for ensemble dynamics were undertaken with the parameters relevant to *E. coli* summarized in **Supplementary file 2**. The surface is discretized into $30 \times 30$ bins.

### 3.4. *Figure 3E* of the main text

Here $z = 4$, $b = 0.5$, and $c = 1$. Periodic boundary conditions in $v$ for the cylinder are assumed. Note that we use the linear relation $\sigma = \alpha(\Delta c)^{-1}$, where $\Delta c$ is the difference of principal curvatures and the value of $\alpha$ is provided in **Supplementary file 2**. Numerical calculations for ensemble dynamics were undertaken with the parameters relevant to *B. subtilis* summarized in **Supplementary file 2**. The bulge is discretized into $20 \times 20$ bins and the cylinder is discretized into $40 \times 40$ bins.

### 3.5. *Figure 3F* of the main text

Here $c = 0.1$, $P = 4$, and $z = \pi$. Periodic boundary conditions in $v$ are assumed. Note that we use the linear relation $\sigma = \alpha(\Delta c)^{-1}$, where $\Delta c$ is the difference of principal curvatures and the value of $\alpha$ is provided in **Supplementary file 2**. Langevin simulations to generate a representative trajectory were undertaken with a single activated filament and the parameters relevant to *E. coli* summarized in **Supplementary file 2**, except $N = 100$. Langevin simulations and numerical calculations for ensemble dynamics were undertaken with the parameters relevant to *E. coli* summarized in **Supplementary file 2**. In the finite case, the surface is discretized into $25 \times 25$ bins into which individual trajectories are aggregated. In the continuum case, the surface is discretized into $200 \times 200$ bins.

### 3.6. *Figure 4A* of the main text

For the ellipsoid, $a = 1$ and $b = 2$. For the torus, $a = 1$, $R = 2$, and $\Phi = 2\pi$. For the helix, $a = 1$, $R = 2$, $\Phi = 2\pi$, $\phi = 1$, and periodic boundary conditions in $v$ are assumed. For all geometries,

Langevin simulations to generate a representative trajectory were undertaken with a single activated filament and $L = 0.4$, $\sigma = 0.3$, and $N = 300$. Numerical calculations for ensemble dynamics were undertaken with the parameters summarized in *Supplementary file 2* but $\sigma = 0.3$, $N$ large enough to correspond to a fixed point for the dynamics ($N = 10^3$), and either $\lambda = 0$ (infinite processivity) or $\lambda = \infty$ (zero processivity). The surfaces are discretized into $30 \times 30$ bins.

### 3.7. *Figure 4B* of the main text

Here $z = 4$, $b = 0.5$, and $c = 1$ or $c = 0.5$. Periodic boundary conditions in $v$ for the cylinders are assumed. Langevin simulations to generate a representative trajectory were undertaken with a single activated filament and $L = 0.4$, $\sigma = 0.3$, and $N = 30$. Numerical calculations for ensemble dynamics were undertaken with the parameters summarized in *Supplementary file 2* but $\sigma = 0.3$, $N$ large enough to correspond to a fixed point for the dynamics ($N = 10^3$), and either $\lambda = 0$ (infinite processivity) or $\lambda = \infty$ (zero processivity). The bulges are discretized into $20 \times 20$ bins and the cylinders are discretized into $40 \times 40$ bins.

### 3.8. *Figure 4C* of the main text

Here $z = 4$ with periodic boundary conditions in $v$. Langevin simulations to generate a representative trajectory were undertaken with a single activated filament and $L = 0.4$, $\sigma = 0.3$, and $N = 15$. Numerical calculations for ensemble dynamics were undertaken with the parameters summarized in *Supplementary file 2* but $\sigma = 0.3$, $N$ large enough to correspond to a fixed point for the dynamics ($N = 10^3$), and either $\lambda = 0$ (infinite processivity) or $\lambda = \infty$ (zero processivity). The surface is discretized into $60 \times 60$ bins.

### 3.9. *Figure 3—figure supplement 1*

The numerical results shown in panel a are identical to those in *Figure 4B* of the main text, with an additional representative trajectory in the case $c = 0.2$. For the simulation in the inset of panel a, $\sigma = 0$. The numerical result shown in panel b is identical to that in *Figure 3E* of the main text.

### 3.10. *Figure 3—figure supplement 3*

The numerical results shown in panel b are identical to those in *Figure 3G* of the main text, with the exception of (1) a constant value of $\sigma = 0.3$ and (2) a quadratic dependence of $\sigma$ on the difference of principal curvatures, $\sigma = \beta(\Delta c)^{-2}$, where the value of $\beta$ is provided in *Supplementary file 2*. A numerical result corresponding to *Figure 3G* of the main text, but with a step size of $L = 2~\mu\mathrm{m}$, is also shown. Here the same parameters summarized in *Supplementary file 2* apply, except the larger value of $L$ implies the following rescaling of parameters: $L = 4$, $N = 6$, and $\lambda = 0.66$. Finally, a numerical result corresponding to *Figure 3G* of the main text, but for a nonzero filament twist of $|\omega_0 a| = 5$ and Gaussian curvature-dependent activation parameter of $\gamma/k_0 = 1$ (see also §2.10 and *Figure 4—figure supplement 4h*) is shown.

### 3.11. *Figure 4—figure supplement 1*

For the torus, $a = 1$, $R = 2$, and $\Phi = 2\pi$. For the helix, $a = 1$, $R = 2$, $\Phi = 2\pi$, and $\phi = 1$ unless otherwise stated, and periodic boundary conditions in $v$ are assumed. Numerical calculations for ensemble dynamics were undertaken with the parameters summarized in *Supplementary file 2* but varying $\sigma$, $N$ large enough to correspond to a fixed point for the dynamics ($N = 10^3$), and either $\lambda = 0$ (infinite processivity) or $\lambda = \infty$ (zero processivity). In panel c, $\sigma$ is fixed at $\sigma = 0$ while $\phi$ varies. The surfaces are discretized into $30 \times 30$ bins.

### 3.12. Figure 4—figure supplement 2

Here $a = 1$ and $b = 2$. Numerical calculations for ensemble dynamics were undertaken with the parameters summarized in **Supplementary file 2** but varying $\sigma$, $N$ large enough to correspond to a fixed point for the dynamics ($N = 10^3$), and either $\lambda = 0$ (infinite processivity) or $\lambda = \infty$ (zero processivity). The ellipsoid is discretized into $30 \times 30$ bins.

### 3.13. Figure 4—figure supplement 3

The numerical results shown in **Figure 4—figure supplement 3a** are identical to those in **Figure 4C** in the main text. The same details apply for **Figure 4—figure supplement 3c**, except that $\sigma$ and $\lambda$ are varied.

### 3.14. Figure 4—figure supplement 4

The numerical results shown in all panels use the parameter values relevant to *E. coli* as summarized in **Supplementary file 2**. Note that we use the linear relation $\sigma = \alpha(\Delta c)^{-1}$, where $\Delta c$ is the difference of principal curvatures and the value of $\alpha$ is provided in **Supplementary file 2**. Generally, $a = 1$, $R = 2$, and $\Phi = 2\pi$ except for panels g and h, for which $R = 10$. All simulated torii were discretized into $30 \times 30$ bins.

