## [Decision Letter]

Thank you for submitting your article "Mechanics and dynamics of translocating MreB filaments on curved membranes" for consideration by *eLife*. Your article has been reviewed by two peer reviewers, and the evaluation has been overseen by a Reviewing Editor and Gisela Storz as the Senior Editor. The reviewers have opted to remain anonymous.

The reviewers have discussed the reviews with one another and the Reviewing Editor has drafted this decision to help you prepare a revised submission.

The manuscript by Wong et al. seeks to link the molecular details of polymer binding and motion on curved surfaces to phenomena observed in the localization and motion of MreB in rod-shaped bacteria. This is timely work and focuses on a multi-scale process that is both fundamental to the way bacteria grow and firmly grounded in molecular biophysics. The authors present a very large number of analytical and numerical results and make a good effort to link these to experimental observations. Of particular note, Gaussian curvature localization (which seems to be somewhat controversial) is linked in this model not to an energetic preference of the proteins for these parts of the cell but rather to a consequence of the motion and the geometry via kinetics.

A series of important changes is requested before the manuscript could be considered for publication in *eLife*.

1) The authors need to do a better and more even-handed job addressing the current theoretical literature that has attempted to explain MreB localization, and comparing their results to previous theoretical works. Specifically, the authors make very different assumptions/focus on different details of the polymers than several previous papers. For example, Quint, Gopinathan and Grason et al., 2016, show that ribbon-like polymers (of which MreB is likely to be one example due to its protofilament architecture) can have both intrinsic curvature and twist which can give rise to localization at specific mean and Gaussian curvatures. Wang and Wingreen, 2013, (which is barely discussed despite being likely the most relevant previous theoretical work on this subject) discuss curvature localization in detail and show that the specific pattern of polymer deformation (e.g. where most of the deformation is focused at the polymer tips) is highly important. Why did the authors of the manuscript under review choose the details of their model and how would it be different if they had added the details considered by others and shown to affect geometric localization? What if the localization or motion did depend on Gaussian curvature or if the filaments were slightly twisted ribbons? The authors specifically state in Appendix 1 "we have assumed the filament to be bent uniformly, but the case of a curvature which varies with position along the filament length can be considered similarly." But would it have made a difference (as suggested by Wang and Wingreen)? Would this fit the data better? Worse? The same but with more parameters? The manuscript is written as if there is no other theoretical work out there and that is just not true. The authors need to both motivate their decisions and compare their results quantitatively to those of previous models in the literature. Otherwise, the reader cannot put the work into context and it is unclear which model is more likely correct given available experimental data.

2) As shown in Figure 2C the model has broader implications in terms of filament proteins and membrane binding than MreB bound to cylindrical membranes and discussion of how it could apply to other biological systems should be included. In particular, the detailed mathematical model and all its generality must be explained in a context that fits the scope of *eLife*. The paper contains notation commonly used in mathematics (but not that common in biophysics). For example, after Equation 3, the angular noise is denoted by N(0, σ^2) where neither N or its arguments are defined nor given biophysical significance. This trend towards a formal mathematical approach is even more prevalent in the supplementary information. Another example, is the discussion in the main text of the processivity parameters λ and *k* in the subsection “Dynamics of translocation”. While they are used in some of the detailed calculations in the supplementary information, the main text with the application to MreB proteins shows results for either zero processivity for which the proteins cannot attain their minimal free energy configuration in regions of high membrane curvature or for infinite processivity, in which case the proteins can move to reside in the high curvature regions of the membrane – with some fluctuations about those positions. Thus, it was not clear why the dynamic calculation was performed in such detail since the main text only showed the results for the two limiting cases of kinetic restrictions and no kinetic restrictions.

3) The kinetic equation (Equation 3) seems to coarse grain over the protein-membrane interactions and intrinsic noise to result in a Langevin equation with an effective noise that depends on the curvatures. It is not at all clear how this is obtained from the standard physical treatment of particle motion where the Langevin equation contains a term corresponding to the physical forces (in this case, due to the curvature energies of the membrane and protein) and another corresponding to white noise (which could also model cell activity if so desired). Had the standard treatment been used, the authors might have been able to predict the steady-state position and orientation as well as the fluctuations about that steady-state using a Boltzmann factor (with an effective noise) which would have provided more intuition than the kinetic simulations that eventually attain such a steady-state.

Specific points:

- The authors state that MreB motion is processive "in contrast to spontaneous or diffusive motion," but then state "we may model the trajectories of filaments as random walks." This seems contradictory.

- "translocates along the largest principal direction." Why was this chosen? Can the authors motivate this with a molecular mechanism via which enzymes/the polymers might know which direction has the largest principle curvature? This to me is the least well motivated part of the model in terms of linking realistic molecular biophysics to the global behavior.

- The supplement is already quite lengthy and detailed, but there remain statements in the text that either state something can be shown, that some parameter/choice is rather unimportant, or simply make an assertion and state "(not shown)". I find this extremely problematic. The authors should back up all of these statements (e.g. with a derivation, calculation, simulation, etc.). This reminded me of the professor who states, "it is trivial to show…". If it is obvious, then it should be easy to show. If it is hard to show, then the off-handed remark seems incorrect.

- "filaments are increasingly misaligned in wider *B. subtilis* cells". Define misaligned so the reader does not have to go to the referenced paper and figure out what this means.

- Figure 3: The '0' and '\infty' designations were confusing for B and D, particularly after reading the fifth paragraph of the subsection “Dynamics of translocation”. I think this is the most confusing part of the text and after reading the text, supplement, and figures/supplementary figures I admit to not fully grasping all of it. Perhaps it could be made more clear.

- Figure 4 inset: Why is this not compared to the experimental data (e.g. from Bratton et al., 2018) as the Mean curvature plot is? This would make this result (a key one for the paper) much stronger.

- Supplementary file 1, 2: Is it possible to put confidence intervals on these estimates from different papers? I understand the authors need to make choices for their modeling, but it would be useful to know how well the field agrees on these numbers.

- Figure 3—figure supplement 1: In the fluorescence image, it is not clear where the cell/bulge are, especially compared to the rest of the figure. Maybe add a schematic?

- Figure 3—figure supplement 2 and Appendix subsection “Cylinder with small wavelength undulations”: The authors state that since Ursell et al. showed a high degree of correlation between mean and Gaussian curvature, that the small wavelength undulating model is better. But in Ursell, this was done for the channel-curved sinusoidal cells and not normal rod cells. Could this make a difference such that the long wavelength model might be better? If it was, would this make a difference?

- Figure 3—figure supplement 3: Unless I missed it, this figure is not used in the text or supplement. Does it matter that the data clearly does not resemble the assumption?

---

## [Author Response]

A series of important changes is requested before the manuscript could be considered for publication in eLife.1) The authors need to do a better and more even-handed job addressing the current theoretical literature that has attempted to explain MreB localization, and comparing their results to previous theoretical works. Specifically, the authors make very different assumptions/focus on different details of the polymers than several previous papers. For example, Quint, Gopinathan and Grason, 2016, show that ribbon-like polymers (of which MreB is likely to be one example due to its protofilament architecture) can have both intrinsic curvature and twist which can give rise to localization at specific mean and Gaussian curvatures. Wang and Wingreen, 2013, (which is barely discussed despite being likely the most relevant previous theoretical work on this subject) discuss curvature localization in detail and show that the specific pattern of polymer deformation (e.g. where most of the deformation is focused at the polymer tips) is highly important.

We thank the reviewers for these comments. We have further explained these previous studies better, and also clarified how our study is different, as follows:

1) In the section on MreB binding, we highlight the work done by Wang and Wingreen in showing that thick, twisted MreB bundles can orient along membranes and have their lengths regulated by twist [Wang and Wingreen, 2013].

2) We also discuss the work by Quint et al. demonstrating that surface regions of negative Gaussian curvature can allow twisted filaments to bind with low elastic energy [Quint et al., 2016].

3) We then detail how our work is different from these studies, in that it includes the directional motion of MreB filaments along their lengths. This directional motion is shown in the new Figure 1—video 1. Specifically, we show how the observed cellular distributions of MreB filaments across different experiments [Hussain et al., 2018; Wong et al., 2017; Kawazura et al., 2017; Ursell et al., 2014; Shi et al., 2017] can be explained with translocation alone. In principle, we have shown that the simplest assumption of MreB motion can give rise to localization.

While we agree with the reviewers that it is interesting to model the combined effects of twist and motion for general filament systems, we note that there is no in vitro evidence that membrane-bound MreB filaments are twisted. Rather, in all experimental observations of membrane-bound filaments, they are flat and untwisted (discussed below). The idea that MreB filaments contain twist is based only on theory, first arising in Wang and Wingreen’s work [Wang and Wingreen, 2013], and later supported by molecular dynamics simulations [Colavin et al., Proc. Natl. Acad. Sci. USA 111, 3585 (2014)]. The only experimental data indicating that MreB filaments might have twist are from observations of in vitro polymerization of *E. coli* MreB in the absence of membranes, where it aggregates into non-physiological bundles [Nurse and Marians, J. Biol. Chem. 288, 3469 (2013)]. These bundles, which appear to have some twist, are likely driven by aggregation of the hydrophobic domains [Esue et al., J. Biol. Chem. 280, 2628 (2005)]. However, when MreB is assembled in the presence of its native membrane substrate, three separate cryo-EM studies have found MreB filaments are flat and untwisted [Salje et al., 2011; van den Ent et al., 2014; Hussain et al., 2018]. When MreB is assembled on flat, supported lipid bilayers, filaments have no twist and are short [Salje et al., 2011; van den Ent et al., 2014]. When MreB filaments are assembled inside liposomes, they become much longer and wrap around the inside of the liposome, but again show no twist.

Regardless, we agree it remains a possibility MreB filaments from different species might be twisted. Thus, we now include in the paper both a more thorough discussion of the previous work, as well as an exploration of how twist affects our model predictions. As shown below in our next response, our model predictions are largely robust to twist.

Why did the authors of the manuscript under review choose the details of their model and how would it be different if they had added the details considered by others and shown to affect geometric localization? What if the localization or motion did depend on Gaussian curvature or if the filaments were slightly twisted ribbons?

We thank the reviewers for this comment. As noted above, we initially chose the parameters of our model based on the untwisted filaments found in all experimental observations of membrane-bound MreB filaments to date. However, we also agree with the reviewers that modeling the different effects of twist and bending rigidity for general filament systems is intriguing. A general model of filament-membrane binding might be applicable for other species of MreB.

Per the reviewers’ suggestion, we have added new work showing the possible effects of three parameters on model predictions: these parameters are the (1) intrinsic filament twist, (2) coupling of filament activation to Gaussian curvature, and (3) filament bending rigidity. A systematic exploration of these parameters has been introduced in a new subsection “Effects of filament twist, flexural rigidity, and Gaussian curvature-dependent activation on model predictions” of the Appendix, Figure 3—figure supplement 3, and Figure 4—figure supplement 4.

Briefly, we summarize our results as follows:

1) Previous work [Quint et al., 2016] has shown that the binding angles of filaments may become biased due to filament twist: Equation 12 of that work,2Csin3⁡θ0cos⁡θ0r2=Kcos⁡(2θ0)2r(sin⁡(2θ0)2r+ω0),predicts the angular deviation from the largest principal direction in a cylinder, θ0, due to twist in the limit that MreB filaments are fully bound to the membrane. Here C is the bending rigidity of a filament, K is the elastic twist stiffness, r is the cell radius, and ω0 is the intrinsic helical twist. While future work should verify the robustness of this equation for nonzero turgor pressures and intrinsic filament curvatures, we found that the corresponding deviation from motion along the largest principal direction does not significantly change our model predictions for *all* possible values of twist on a torus.

2) We then examined the influence of Gaussian curvature-coupling on our model predictions. While Quint et al. show that twist may contribute to Gaussian curvature-dependent activation, the coupling of filament activation to Gaussian curvature can, in general, be influenced by other cellular processes and may be independent from twist. We therefore varied the coupling parameter independently. We found that, for a broad range of such couplings, the model predictions were quantitatively similar, with larger localization at the inner edge of a torus for larger Gaussian curvature couplings. Nevertheless, our prior experiments measuring MreB fluorescence intensity in cells confined to curved microchambers suggest the magnitude of Gaussian curvature coupling, if it exists, to be small [Wong et al., 2017], and our measurements in that work can be explained without Gaussian curvature coupling (Figure 4—figure supplement 4h). We have also shown that the results pertaining to a different geometry, that of a cylinder with small wavelength undulations, are largely robust to twist and Gaussian curvature coupling in Figure 3—figure supplement 3.

3) Finally, we showed that our results are largely robust to MreB filament rigidity by modeling both the binding and motion of MreB filaments with significantly smaller and larger bending rigidities. While the details of membrane binding differ among these cases, the ensuing motion and predictions of localization do not. The main difference we observed is that MreB bundles are not predicted to bend to conform to the membrane; rather, the membrane conforms to the bundle even for physiological values of turgor in *E. coli*. This is qualitatively consistent with Wang and Wingreen’s model, which predicts the largest deformation to occur at the tips [Wang and Wingreen, 2013].

To clarify these points in the main text, a discussion of the works mentioned above has been added to the first paragraph of the subsection “Mechanics of binding”, where we further detail our rationale for focusing on specific parameter values. We state that our results remain largely robust to twist, Gaussian curvature coupling, and filament rigidity in a new subsection “Dependence of localization on processivity and Gaussian curvature”, where we refer readers to the Appendix subsection “Effects of filament twist, flexural rigidity, and Gaussian curvature-dependent activation on model predictions” for a more detailed discussion.

The authors specifically state in Appendix 1 "we have assumed the filament to be bent uniformly, but the case of a curvature which varies with position along the filament length can be considered similarly." But would it have made a difference (as suggested by Wang and Wingreen)? Would this fit the data better? Worse? The same but with more parameters?

As noted above in our first response to point (1), Wang and Wingreen model MreB bundles that have a significantly larger bending rigidity. While it is interesting to explore the relation between different bundle shapes and their accompanying membrane deformations (see also our second response to point (1)), we note the following. First, these large MreB bundles are not physiological, as they are only observed in the absence of membranes (as discussed further in response to point (1)). Second, we note that all three in vitro cryo-EM studies of MreB have observed membrane-bound MreB filaments to be bent uniformly along their lengths [Hussain et al., 2018; Salje et al., 2011; van den Ent et al., 2014]. This is also a natural result if MreB filaments deform against the membrane and are short, so that the membrane curvature can be assumed uniform. Third, we note that the goal of our work is different from that of Wang and Wingreen’s: we were less interested in exploring the different ways in which thick bundles can deform membranes and more focused on clarifying why MreB filaments (1) align along principal curvatures and (2) localize differently to different cellular regions.

Furthermore, for the case of MreB *filaments –* the case on which we focus in this work – a calculation based on Equation S2 in the Appendix shows that varying curvature with position along the filament will result in the same predictions of alignment along directions of largest curvatures, given that the filaments are, on average, more curved than the membrane. In particular, let κs⁢(ℓ) denote the intrinsic filament curvature as a function of position along the filament length, ℓ, and κ denote the deformed curvature. In cylindrical cells, κ does not vary as a function of ℓ because it is most energetically favorable for the filament to bend completely to match the ambient membrane curvature, as we have shown in our work. The total bending energy of the filament is thenEbend=B2⁢∫0Lf(κ-κs⁢(ℓ))2⁢𝑑ℓ.where B is the bending rigidity (identical to the variable C in Quint et al.’s work mentioned above) and Lf is the filament length. When binding to an angle that deviates from the circumferential direction in a cylinder, the deformed curvature will be smaller: let κ=κ0-κ′, where κ0 is the curvature along the circumferential direction of a cylinder and κ′≥0 is a constant correction to κ0 depending on the binding angle. Then, the difference in bending energies between binding in the direction of κ as opposed the circumferential κ0 direction isB2⁢∫0Lf(κ′)2⁢𝑑ℓ+2⁢κ′⁢B2⁢∫0Lf(κs⁢(ℓ)-κ0)⁢𝑑ℓ,which is larger than zero provided the filament is, on average, more curved than the cell (that is, the second term above is non-negative). Hence, because the binding orientation is robust, our model predictions will remain the same.

As readers may be interested in calculations similar to the above, we have appended the foregoing discussion to the Appendix subsection “Preferred orientation of filament binding and binding phase diagram”. We have, however, decided to keep matters simple in the main text by focusing on the case of a uniformly bent MreB filament.

The manuscript is written as if there is no other theoretical work out there and that is just not true. The authors need to both motivate their decisions and compare their results quantitatively to those of previous models in the literature. Otherwise, the reader cannot put the work into context and it is unclear which model is more likely correct given available experimental data.

We appreciate the reviewers’ note and have made a serious attempt to address it, as detailed in response to point (1). Specifically, we have discussed the two theoretical studies of Quint et al. and Wang and Wingreen in the first paragraph of the subsection “Mechanics of binding” and “Dependence of localization on processivity and Gaussian curvature” in the main text, where we also detail the rationale behind the parameter choices in this work (see also the response to the seventh Specific point). As it remains possible that MreB from different species may be twisted and thick, we have shown that our results are largely robust to filament twist, rigidity, and Gaussian curvature coupling in a new subsection “Effects of filament twist, flexural rigidity, and Gaussian curvature-dependent activation on model predictions”, of the Appendix. We emphasize the main difference that indeed, we know of no prior work on similar curvature-based, directed random walks used to model MreB motion in this work.

2) As shown in Figure 2C the model has broader implications in terms of filament proteins and membrane binding than MreB bound to cylindrical membranes and discussion of how it could apply to other biological systems should be included. In particular, the detailed mathematical model and all its generality must be explained in a context that fits the scope of eLife.

We appreciate the reviewers’ insight that our results can be cast in a broader context. To address this point, we have substantially revised the main text to make it more broadly applicable as follows:

1) We better highlight previous theoretical work and discuss how our approach is different from these studies.

2) We have reorganized the Results to introduce a new subsection “Dependence of localization on processivity and Gaussian curvature”, which explores our model in generality and discusses the applicability of our results to different filament systems. In particular, we have extended our model to explore the effects of twist, Gaussian curvature coupling, and different filament bending rigidities, as detailed in our second response to point (1).

3) Finally, we have substantially rewritten the Discussion as to discuss potential applications of our results to other contexts, including bacterial cytokinesis and membrane trafficking, growth, and movement in both prokaryotes and eukaryotes.

We hope that these changes better highlight the broad scope of our work.

The paper contains notation commonly used in mathematics (but not that common in biophysics). For example, after Equation 3, the angular noise is denoted by N(0, σ^2) where neither N or its arguments are defined nor given biophysical significance. This trend towards a formal mathematical approach is even more prevalent in the supplementary information.

We have clarified the meaning of the notation in the main text and, whenever possible, in the Appendix. This includes the following changes:

1) The offending phrase and its surrounding context has been edited to read “Here and below, we assume ξn∼N(0,σ2)—that is, the angular noise is normally-distributed, with mean zero and a variance, σ2, to be inferred from data—and note that the *translocation noise*, σ, may depend on quantities such as the principal curvatures, as discussed later”.

2) “where (x,y)∈R2” has been changed to “where x and y are real numbers”.

3) The meaning of the notation ∇h is clarified as “the gradient of h”.

4) The phrase “r=r(u,v)⊂R3” has been clarified. The phrase and its surrounding context now reads “We consider the membrane as a parametric surface, r=r(u,v), embedded in three-dimensional space (R3) with surface coordinates u and v”.

5) The notation O[(∇h)2] has been clarified as “where the big-O notation signifies*|H−12∇2h|≤M(∇h)2*when 0<(∇h)2<δ for some positive numbers δ and M”.

6) The notation ∂Ω has been clarified as “the boundary of Ω”.

7) The notation ξn∼N(0,σ2) is again clarified in the Appendix. The revised sentence reads “Here and below, ξn∼N(0,σ2)—that is, the angular noise is normally-distributed, with mean *0* and variance σ2*”*.

We thank the reviewers for pointing out this issue and hope that these clarifications will make our work more accessible to a non-mathematically inclined audience.

Another example, is the discussion in the main text of the processivity parameters *λ* and *k* in the subsection “Dynamics of translocation”. While they are used in some of the detailed calculations in the supplementary information, the main text with the application to MreB proteins shows results for either zero processivity for which the proteins cannot attain their minimal free energy configuration in regions of high membrane curvature or for infinite processivity, in which case the proteins can move to reside in the high curvature regions of the membrane – with some fluctuations about those positions. Thus, it was not clear why the dynamic calculation was performed in such detail since the main text only showed the results for the two limiting cases of kinetic restrictions and no kinetic restrictions.

We apologize for the confusion. The cases of zero and infinite processivity were explored to demonstrate limiting cases, as to first cast our results in a broader context and understand the underlying physics. To better motivate this exploration, we have significantly reorganized our Results as to first discuss the implications of our model to MreB (subsection “Implications to MreB localization”) and then explore our model in a broader context (subsection “Dependence of localization on processivity and Gaussian curvature”). The results for MreB, which involve a finite processivity (Supplementary file 2) and are neither in the limit of zero nor infinite processivity, are now shown in Figure 3. The subsequent exploration of the physics is now shown in Figure 4.

To further clarify the relevance of finite processivity to MreB, we have:

1) Appended a clarifying sentence to the legend of Figure 3: “In this figure, the localization of MreB filaments, which exhibit a finite processivity and are assumed to follow the parameter values summarized in Supplementary file 2, is shown,”

2) Revised a relevant sentence in the legend of Figure 4 to read “Note that cases of zero processivity correspond to uniform distributions and that we have considered the limiting cases of zero and infinite processivity here. Figure 3 shows numerical results for the case of a large, but finite, processivity relevant to MreB,” and changed relevant phrases in the main text to better emphasize the MreB-specific (Figure 3) and general (Figure 4) processivity values adopted, including “While we assume the values of λ and σ to be based on these measurements, we examine the effects of varying λ and σ in the following section” and “in the case of small λ corresponding to large processivity – a limiting case that is relevant to MreB”.

We hope this clarification also better motivates some of the calculations undertaken in the Appendix.

Finally, we note that processivity plays an important role in our model: in the limit of zero processivity, MreB filaments are assumed to be deactivated as soon as they are activated. If the activation of MreB filaments were simply proportional to area, as we have assumed in our work, then the binding orientation is inconsequential and no localization can occur, as shown in Figure 4. However, we would also like to emphasize that, at least for the model considered in our work, there is generally no explicit connection between processivity and (1) attaining a minimal free energy configuration or (2) moving to reside in *highly curved* membrane regions in our work. For (1), we hypothesize the minimal free energy configuration to be relevant for binding and orienting the filament only (subsection “Mechanics of binding”). The processivity is a variable that determines the lifetime of MreB filaments while moving and is not necessarily related to binding. As for (2), the movement of MreB filaments along the largest principal directions need not entail that they will end up residing in highly curved membrane regions. Whether or not such correlation manifests depends on the geometry at hand, as detailed more carefully in the new subsection “Dependence of localization on processivity and Gaussian curvature”. In other words, localization is not generically correlated with scalar measures of curvature. Figure 4C illustrates this point succinctly: in this geometry, motion along the largest principal directions results in accumulation of filaments, but the Gaussian curvature vanishes everywhere. In contrast, correlations between filament concentration and the Gaussian curvature exist for the geometry shown in Figure 3F.

3) The kinetic equation (Equation 3) seems to coarse grain over the protein-membrane interactions and intrinsic noise to result in a Langevin equation with an effective noise that depends on the curvatures. It is not at all clear how this is obtained from the standard physical treatment of particle motion where the Langevin equation contains a term corresponding to the physical forces (in this case, due to the curvature energies of the membrane and protein) and another corresponding to white noise (which could also model cell activity if so desired). Had the standard treatment been used, the authors might have been able to predict the steady-state position and orientation as well as the fluctuations about that steady-state using a Boltzmann factor (with an effective noise) which would have provided more intuition than the kinetic simulations that eventually attain such a steady-state.

We thank the reviewers for this comment. The main point raised is the justification of using kinetic simulations that attain a steady state but do not relate directly to a Langevin equation describing an equilibrium process, whose steady state corresponds to the Boltzmann distribution. A crucial difference between our work and such an approach is that the dynamics here are inherently away from equilibrium, which implies that the steady-state solution will not generally be well described by the Boltzmann distribution. The non-equilibrium nature of the dynamics can be appreciated by observing that detailed balance is broken due to directional motion. To see this for the cylinder, we may consider the transitions in (*X_n_, χ_n_*)-space between two small area elements as below; as in the main text, *X_n_* and *χ_n_* respectively denote the position and direction of the filament at a timestep n.

**Author response image 1. respfig1:** Broken detailed balance in translocation dynamics on a cylinder. Here two area elements are separated by a distance *L* corresponding to the translocation step size. The red arrows indicate preferred translocation directions – that is, principal curvature directions oriented along the sign of *χ_n_*. The corresponding dynamics (right) can be written, where *k_α_* denotes the kinetic rate corresponding to translocation along a deviatory angle of α from the preferred translocation direction. For small α (α < 90°), the forward and backward rates are unequal because the probability of a filament reorienting to an angle of α+180° will be smaller than that of an angle α.

Note that the steady-state probability of a filament in (*X_n_*, *χ_n_*)-space is uniform: it is uniform in position space due to symmetry and in direction space due to the hypothesis that *χ_n_* is uniformly random (but does not vary in *n*). However, detailed balance does not hold in Author response image 1 because, for small deviatory angles between the two area elements, the transition rate corresponding to motion in the direction of *χ_n_* is larger than the transition rate corresponding to motion in the direction of *−χ_n_*. Hence, the dynamics is out of equilibrium. This argument applies across many of the simulations considered in our work. Intriguingly, the non-equilibrium dynamics implies the expenditure of energy, and prior studies have shown that MreB filaments require the energy-intensive process of cell wall synthesis in order to move processively [Garner et al., 2011; van Teeffelen et al., 2011; Domínguez-Escobar et al., 2011].

More broadly, while we have been able to obtain theoretical predictions for displacements and fluctuations in the simple case of an infinite cylinder (subsection “Cylinder” of the Appendix) and infer the filament concentrations for other limited geometries (subsections “Torus and Helix”, “Ellipsoid”, and “Filament concentration is independent of Gaussian curvature” of the Appendix), we have not been able to recast Equation 3 into a more accessible form that, even away from equilibrium, would permit a general solution. We therefore thought it best to describe the stochastic process of Equation 3 accessibly and resort to numerical simulation. We hope that this will provide a foundation for further theoretical work which builds on our model.

Specific points:- The authors state that MreB motion is processive "in contrast to spontaneous or diffusive motion," but then state "we may model the trajectories of filaments as random walks." This seems contradictory.

We understand the reviewer’s confusion, as the motion of MreB filaments is processive and highly directional. We use the term “random walks” to designate any stochastic process which describes a path comprising a succession of random steps. To mitigate confusion, we now use the term “biased random walks” and clarify, in the following sentence, that “a ‘biased random walk’ refers to a succession of random steps which may be processive: while the mean-squared displacement of a filament will be approximately quadratic, and not linear, in time, the processive motion we consider is random only because the translocation direction can deviate from directions of largest membrane curvature due to sources of stochasticity”.

- "translocates along the largest principal direction." Why was this chosen? Can the authors motivate this with a molecular mechanism via which enzymes/the polymers might know which direction has the largest principle curvature? This to me is the least well motivated part of the model in terms of linking realistic molecular biophysics to the global behavior.

The oriented motion of MreB filaments has been well-documented in both *E. coli* and *B. subtilis*. In cells, MreB filaments move directionally, and this motion is oriented predominantly around the cell width and perpendicular to the long axis [Garner et al., 2011; van Teeffelen et al., 2011; Domínguez-Escobar et al., 2011]. In our original work [Hussain et al., 2018], we worked to understand what caused MreB motion to be circumferentially oriented. We found that MreB orientation is most likely determined only by MreB filaments binding to membranes. The orientation arises as these highly inwardly-curved filaments bind to the membrane in a manner which maximizes their interaction energy – i.e., filaments orient along the direction of largest curvature. We demonstrated that MreB translocates along the direction of largest curvature in a variety of cell shapes, including rods, round cells, and bulged cells. Furthermore, we showed that MreB filaments also orient in directions determined by their binding orientations in both spheroplasted cells and liposomes (Author response image 2), in agreement with previous work [Olshausen et al., 2013]. In this work, we now explore the implications of filament orientation and directional motion on the cellular distribution of filaments.

**Author response image 2. respfig2:** In *B. subtilis*, the angular distribution of membrane-bound filaments within protoplasts confined to be rod-shape (*protoplast filaments*) is peaked at 90° relative to the long axis (mean deviation = 34°, *n* = 147), similar to that of MreB motion in wide (TagO-depleted), confined cells (*mother machine tracks*) (mean deviation = 36°, *n* = 359) and MreB motion in wild-type cells (*Wt tracks*) (mean deviation = 34°, *n* = 1041). This figure is reproduced from Figure 3F of [Hussain et al., 2018] and adapted for the new Figure 1C of this work.

As it appears this background was not evident in our initial draft, in the updated manuscript we have summarized the evidence for principal curvature-based motion in a new Figure 1C and the section on Dynamics of translocation. We hope this improved discussion of previous work clarifies the rationale for our model.

If the reviewers are querying in regards to how the directional motion of MreB filaments arises, we note that the exact mechanism is not known, although this motion requires – and is perhaps driven by – cell wall synthesis [Garner et al., 2011; van Teeffelen et al., 2011; Domínguez-Escobar et al., 2011]. We believe many possibilities exist, but have decided not to speculate upon them in the text.

- The supplement is already quite lengthy and detailed, but there remain statements in the text that either state something can be shown, that some parameter/choice is rather unimportant, or simply make an assertion and state "(not shown)". I find this extremely problematic. The authors should back up all of these statements (e.g. with a derivation, calculation, simulation, etc.). This reminded me of the professor who states, "it is trivial to show…". If it is obvious, then it should be easy to show. If it is hard to show, then the off-handed remark seems incorrect.

We thank the reviewers for this comment and have sought to address it by backing up all such statements as follows:

1) In the main text, we have replaced the only instance of the phrase “it can be shown” before “CF is uniform over the surface of an ellipsoid” with a reference to the Appendix, which indeed shows this via both calculations and simulations (subsection “Ellipsoid” of the Appendix and Figure 4—figure supplement 2).

2) In subsection “Model of a protein filament binding to a membrane” of the Appendix, we have removed the sentence “For the parameter values summarized in Supplementary file 2, it can be shown that *l_b_*=*L_f_* below,” as it is not necessary to claim this. We have instead mentioned in subsection “Finite element solutions of the shape equation” of the Appendix, that the numerical optimization runs over varying sizes of *U*, which corresponds to the bound length of the filament, and that they predict *l_b_*=*L_f_*. This result is consistent with the linear dependence on *l_b_* of the free energy change due to binding in the case of physiological turgor pressures, as we have clarified following Equation S11 of the Appendix.

3) The claim that “our results do not significantly change for different *b*,” the fractional cross-sectional binding region, has been supplemented with a reference to Figure 4 of [Hussain et al., 2018] which shows this in panels F and G.

4) In subsections “Torus and Helix” and “Ellipsoid” of the Appendix, claims that certain results hold in the limit of large noise (σ→∞) have been corroborated by showing the results of additional simulations in this limit in Figure 4—figure supplements 1 and 2.

5) In subsection “Cylinder with a bulge” of the Appendix, the claim that bulges attract filaments in the case of no noise and large processivity is represented by a simulation which shows this (inset of Figure 3—figure supplement 1A).

6) In subsection "Effects of principal curvature-dependent translocation noise and varying filament step size on model predictions”, claims that varying the dependence of σ on ∆*c* do not significantly change our results are corroborated with new simulation results in a revised Figure 3—figure supplement 3 (see also the response to the last Specific point).

We believe these revisions address all instances of unsupported assertions, both in the main text and in the Appendix.

- "filaments are increasingly misaligned in wider B. subtilis cells". Define misaligned so the reader does not have to go to the referenced paper and figure out what this means.

We thank the reviewers for drawing our attention to this point. We have revised the offending phrase and its surrounding context to read “As the distribution of MreB filament angles gradually becomes broader as *B. subtilis* cells become wider” and hope this clarifies our meaning.

- Figure 3: The '0' and '\infty' designations were confusing for B and D, particularly after reading the fifth paragraph of the subsection “Dynamics of translocation”. I think this is the most confusing part of the text and after reading the text, supplement, and figures/supplementary figures I admit to not fully grasping all of it. Perhaps it could be made more clear.

We thank the reviewers for this comment and hope that they find this point clarified in our third response to point (2). To clarify the definition of processivity, we have added the phrase “that is, the mean number of steps that a filament takes on the membrane surface before becoming inactive” upon first mention of processivity.

- Figure 4 inset: Why is this not compared to the experimental data (e.g. from Bratton et al., 2018) as the Mean curvature plot is? This would make this result (a key one for the paper) much stronger.

We thank the reviewers for pointing this out and have split the old Figure 4E into two panels, which are now shown in Figure 3G. Here, we individually compare our predictions with statistics from the works of Ursell et al. and Shi et al. [Ursell et al., 2014; Shi et al., 2017] for the mean curvature and Bratton et al. [Bratton et al., 2018] for the Gaussian curvature.

While the geometry we have assumed in Figure 3F exhibits a larger range of mean and Gaussian curvatures than is needed to explain both the mean and Gaussian curvature data, we note that Bratton et al.’s measurements of filament enrichment in wild-type *E. coli* cells are quantitatively consistent with our predictions and indicates a decreased concentration of MreB filaments at regions of positive Gaussian curvature. We therefore believe that the revised Figure 3G strengthens our comparison to previous work.

- Supplementary file 1, 2: Is it possible to put confidence intervals on these estimates from different papers? I understand the authors need to make choices for their modeling, but it would be useful to know how well the field agrees on these numbers.

We appreciate the reviewers’ point. To address it, we have performed a detailed literature search and appended different estimates of parameters in Supplementary files 1 and 2, along with relevant references. We hope that this additional information better contextualizes the values chosen.

- Figure 3—figure supplement 1: In the fluorescence image, it is not clear where the cell/bulge are, especially compared to the rest of the figure. Maybe add a schematic?

We thank the reviewers for pointing this out and have added a schematic to Figure 3—figure supplement 1.

- Figure 3—figure supplement 2 and Appendix 1 subsection “Cylinder with small wavelength undulations”: The authors state that since Ursell et al. showed a high degree of correlation between mean and Gaussian curvature, that the small wavelength undulating model is better. But in Ursell, this was done for the channel-curved sinusoidal cells and not normal rod cells. Could this make a difference such that the long wavelength model might be better? If it was, would this make a difference?

We thank the reviewers for this comment. The long wavelength model cannot make a difference for an additional reason: the ranges of Gaussian and mean curvatures predicted by such a model would be significantly smaller than those measured for wild-type, unconfined, sinusoidally-confined, thin, and wide *E. coli* cells from the works of Ursell et al., Shi et al., and Bratton et al. [Ursell et al., 2014; Shi et al., 2017; Bratton et al., 2018]. For instance, the range of mean curvature predicted by the long wavelength model in Figure 3—figure supplement 2 is, in dimensionful units, [.994 μm^-1^, 1.012 μm^-1^]. The data for both confined and unconfined cells from Ursell et al.exhibit a much larger range of approximately [0 μm^-1^, 2 μm^-1^]. Thus, strong regions of curvature are necessary to recapitulate the experimental data.

To clarify this point further, we have introduced additional text to the subsection “Cylinder with small wavelength undulations” of the Appendix, and a relevant sentence now reads: “However, the Gaussian and mean curvatures are uncorrelated in this case, inconsistent with the positive correlation observed in [Ursell et al., 2014] for cells growing in sinusoidally-shaped chambers, and the ranges of Gaussian and mean curvatures are significantly smaller than those measured for wild-type, unconfined, sinusoidally-confined, thin, and wide cells in different experiments (Figure 3—figure supplement 3A and Figure 3G of the main text).” We hope that this better motivates our consideration of a short wavelength geometry.

- Figure 3—figure supplement 3: Unless I missed it, this figure is not used in the text or supplement. Does it matter that the data clearly does not resemble the assumption?

Figure 3—figure supplement 3 is referenced in subsection “Effects of principal curvature-dependent translocation noise and varying filament step size on model predictions” of the Appendix, in addressing the possibility that the translocation noise, σ, depends on the difference of principal curvatures, ∆*c* (subsection “Implications to MreB localization”, first paragraph in the main text). The figure shows the empirically observed dependence of σon cell width, along with a linear fit. While the linear fit clearly does not capture all the variability in the data, the point is that similar dependencies (including the quadratic fit below) of σ on ∆*c* do not qualitatively change our results. To show this explicitly (and in the spirit of our response to the third Specific point), we have regenerated the theoretical predictions of Figure 3G, one of the main results of this work, for different fits of σ as a function of∆*c*. As discussed in response to points and (3), we have also done the same for different values of filament twist and Gaussian curvature coupling. For all cases, we have added new panels to Figure 3—figure supplement 3 showing the qualitative robustness: all assumptions still result in filament enhancement at regions of negative Gaussian curvature.

We thank the reviewers for giving us the opportunity to clarify this point.